# Molecular Umbrella as a Nanocarrier for Antifungals

**DOI:** 10.3390/molecules26185475

**Published:** 2021-09-08

**Authors:** Andrzej S. Skwarecki, Dorota Martynow, Maria J. Milewska, Sławomir Milewski

**Affiliations:** 1Department of Pharmaceutical Technology and Biochemistry and BioTechMed Centre, Gdańsk University of Technology, 80-233 Gdańsk, Poland; andrzej.skwarecki1@pg.edu.pl (A.S.S.); dorota.koperkiewicz@gmail.com (D.M.); 2Department of Organic Chemistry and BioTechMed Centre, Gdańsk University of Technology, 80-233 Gdańsk, Poland; marmilew@pg.edu.pl

**Keywords:** molecular umbrella, antifungals, conjugates

## Abstract

A molecular umbrella composed of two *O*-sulfated cholic acid residues was applied for the construction of conjugates with cispentacin, containing a “trimethyl lock” (TML) or *o*-dithiobenzylcarbamoyl moiety as a cleavable linker. Three out of five conjugates demonstrated antifungal in vitro activity against *C. albicans* and *C. glabrata* but not against *C. krusei*, with MIC_90_ values in the 0.22–0.99 mM range and were not hemolytic. Antifungal activity of the most active conjugate **24c**, containing the TML–pimelate linker, was comparable to that of intact cispentacin. A structural analogue of **24c**, containing the Nap-NH_2_ fluorescent probe, was accumulated in *Candida* cells, and TML-containing conjugates were cleaved in cell-free extract of *C. albicans* cells. These results suggest that a molecular umbrella can be successfully applied as a nanocarrier for the construction of cleavable antifungal conjugates.

## 1. Introduction

In the present SARS COVID-19 pandemic period, it is obvious that in the field of infectious diseases, special attention is focused on progress in the prevention of viral infections and antiviral chemotherapy. However, one cannot underestimate the fact that microbial resistance to antibiotics has emerged, and its spread worldwide has resulted in another significant threat to public health [1]. This challenge is also valid in the case of infections caused by human pathogenic fungi. Fungal micro-organisms from the *Candida* genus, especially *Candida albicans* and *Candida glabrata* but also an emerging pathogen *Candida auris* and filamentous fungi of *Aspergillus* spp., are etiological factors of many serious, often deadly infectious diseases, especially in immunocompromised patients [2]. *C. albicans* is considered the fourth most popular etiological agent of nosocomial infections worldwide [3]. There are several mechanisms of drug resistance of human pathogenic fungi, including those concerning alterations of drug transport across the microbial membranes, resulting from an impaired uptake of a drug molecule by membrane-located permeases or from its efflux, driven by multi-drug (MDR) energy-dependent drug transporters [4]. There is a generally accepted opinion on the urgent need for novel antifungal chemotherapeutics, especially these attacking new molecular targets. However, a number of drug candidates, including known inhibitors of intracellular enzymes identified as targets for antifungals, exhibit poor cellular penetration, owing to their hydrophilicity. One of the most promising strategies for overcoming this problem is the use of molecular carriers to ensure the delivery of enzymatic inhibitors to the intracellular target site [5], which is known as the “Trojan horse strategy”. This approach is based on idea of conjugation of an active substance with a macromolecular or low molecular weight organic carrier easily penetrating the cell membrane [6]. After internalization, the conjugate remains intact or is cleaved, and the released active component can reach its intracellular target. Among the different organic nanocarriers proposed so far, an unique mechanism of internalization is characteristic for compounds known as “molecular umbrellas” [7]. They are “amphomorphic” compounds composed of two or more facial amphiphiles (mostly bile acids) that are connected to a central scaffold (spermine or spermidine). Molecular umbrellas were found capable of transporting certain hydrophilic molecules across liposomal and biological membranes [8] but have not been used so far as nanocarriers in antimicrobial conjugates. Interestingly, the conjugation of a molecular umbrella with known antifungal antibiotic Amphotericin B results in the “taming” of this drug, and as a consequence, improved selective toxicity [9,10]. 

In this work, rationally designed cleavable conjugates of the diwalled molecular umbrella with cispentacin, an inhibitor of Ile-tRNA^Ile^ synthetase, have been tested as antifungal agents. 

## 2. Results

### 2.1. Rationale for Design of Conjugates

A basic rationale in this work was application of the molecular umbrella scaffold for the construction of conjugates with an inhibitor of intracellular fungal enzyme. The molecular umbrella was supposed to play the role of a nanocarrier responsible for the translocation of a conjugate through the fungal cytoplasmic membrane. Then, an enzymatic inhibitor should have been released from a conjugate intracellularly due to the presence of a cleavable linker, joining the nanocarrier and the cargo. 

The simplest diwalled molecular umbrella was chosen as an optimal carrier, since it had been previously shown as the most effective umbrella component of conjugates with an antifungal antibiotic Amphotericin B, where its presence resulted in decreased mammalian toxicity [10]. We expected that the larger, tetra- or octawalled umbrellas may have problems with penetration through the fungal cell wall. Cispentacin, chosen as a cargo component, is an inhibitor of fungal Ile-tRNA^Ile^ synthetase and as a consequence protein biosynthesis [11], which demonstrates antifungal activity [12]. This compound, for its polar character, does not penetrate biological membranes by simple diffusion but is taken up by the transmembrane transporter, proline permease [13]. For this reason, its activity is lower in L-proline containing complex media, where this amino acid competes with cispentacin for the permease and fungal resistance to this antifungal, and its analogue BAY 10-8888 is most often caused by decreased accumulation [14]. The need for a cleavable linker was obvious, since cispentacin interacts with its target exclusively as an intact molecule. In this work, two types of such linkers were employed. *o*-Dithiobenzylcarbamoyl moiety, previously successfully applied in the synthesis of a molecular umbrella–oligopeptide conjugate [8], contains a disulfide bond, which is potentially cleavable upon an intracellular thiol exchange reaction with the reduced glutathione. Nevertheless, this linker has not been employed so far for the construction of antimicrobial umbrella conjugates. The esterase-triggered TML (‘trimethyl lock’) [15], applied originally in antibacterial siderophore conjugates [16,17], has been for the first time used for the construction of potential antifungal conjugates.

### 2.2. Chemistry

Nine cleavable conjugates of a diwalled molecular umbrella with an antifungal agent, cispentacin, or with a fluorescent probe, Lys(Mca) (4-carboxymethyl-7-methoxycoumarin) or Nap-NH_2_ (*N*-butyl-4-aminoethyl-1,8-naphtalimide), as cargo components were synthesized (Figure 1). Seven out of nine conjugates contained the esterase-sensitive TML system, and the remaining two contained the *o*-dithiobenzylcarbamoyl linker. 

The synthesis of the esterase-labile TML linker started from pimelic **2a** and suberic **2b** acids, which were converted into corresponding cyclic anhydrides, using acetic anhydride. The resulting anhydrides **1b**,**c** and commercially available succinic anhydride **1a** were subsequently lysed with *tert*-butanol to dicarboxylic acid monoesters **3a**–**c** in the manner described by Cisneros et al. [18].

The *O*-acylated TML systems **14a**–**c** were prepared in a multistep synthesis presented in Figure 1, being a modified version of the procedure described by Ji and Miller [16]. Condensation of the commercially available methyl 3-methylbut-2-enoate **4** with 3,5-dimethylphenol **5** gave lactone **6**, which after reduction with LiAlH_4_ resulted in the primary alcohol **7**. Subsequently, the phenolic hydroxyl group of **7** was selectively protected with TBDMSCl (*tert*-butyldimethylchlorosilane). The protected alcohol **8** was acylated with appropriate carboxylic acid **3a**–**c** by Steglich esterification. The primary hydroxyl groups of esters **9a**–**c** were deprotected under mildly acidic conditions, resulting in alcohols **10a**–**c**, which underwent subsequent oxidation to corresponding aldehydes **11a**–**c** and then to carboxylic acids **12a**–**c**. The formed carboxylic group was protected with benzyl bromide, thus forming triesters **13a**–**c**. The subsequent deprotection of *tert*-butyl esters with trifluoroacetic acid gave carboxylic acids **14a**–**c** (Figure 1).

Formation of the molecular umbrella structure on the prepared TML linker started with the conjugation of carboxylic acids **14a**–**c** with Boc-protected spermidine **16**, as presented in Figure 2. The synthesis of protected spermidine was accomplished in a one-step reaction of spermidine **15** with 2 equivalents of protecting agent, Boc-ON. The resulting amine **16** was subsequently acylated with NHS (*N*-hydroxysuccinimide)-activated TML linkers **17a**–**c** (NHS/DCC method), and then, the benzyl protecting group was removed by hydrogenolysis on Pd/C catalyst. The resulting carboxylic acids **19a**–**c** were universal building blocks for further conjugation with cargo molecules.

The antifungal agent, cispentacin, was connected to the ‘trimethyl lock’ linker by amide bond formation, which was accomplished by the activation of carboxylic acids **19a**–**c** with TDBTU (*O*-(3,4-dihydro-4-oxo-1,2,3-benzotriazin-3-yl)-*N*,*N*,*N*′,*N*′-tetramethyluronium tetrafluoroborate) followed by aminolysis with cispentacin (Figure 3). The resulting amides **23a**–**c** were deprotected under acidic conditions of trifluoroacetic acid and used for the ultimate formation of the molecular umbrella. For that purpose, deoxycholic acid **20a** and cholic acid **22b** were converted into appropriate active esters **22a**–**c**. The formation of ester **22c** was straightforward and was accomplished using the NHS/DCC technique. For the preparation of active esters **22a**,**b**, cholic and deoxycholic acids were previously converted into sulfate derivatives **21a**,**b** by esterification with a sulfur trioxide/pyridine complex. Then, salts **21a**,**b** were converted to active esters **22a**,**b** with TDBTU (Figure 3). Active esters **22a**,**b** were used for the acylation of spermidine terminal amino groups, which resulted in final molecular umbrella–cispentacin conjugates **24a**–**d**, differing by the length of the “trimethyl lock” containing linkers and the presence or absence of sulfate groups in the cholic acid “walls”.

Conjugates carrying fluorescent probes, Lys(Mca) or Nap-NH_2_, as cargo were prepared analogously as conjugates **24b**–**d**. In the case of conjugate **26a**, an active ester **22c** was used as an acylating agent (Figure 4). 

The synthesis of molecular umbrella conjugates with a *o*-dithiobenzoylcarbamoyl linker was inspired by the research work of Jing et al. [19]. As shown in Figure 5, the synthesis began with commercially available active diester **28**, which was converted to a dimeric structure **29** by amide formation with 2 equivalents of Boc_2_-protected spermidine **16**. Intermediate **29** was subsequently cleaved to corresponding thiol **30**, using TCEP (*tris*(2-carboxyethyl)phosphine) as a reducing agent. Then, a thiol group was activated with asymmetric disulfide **31**, affording a mixed disulfide **32**, which underwent an activation reaction with DSC (*N*,*N*′-disuccinimidyl carbonate). Active ester **33** reacted with cispentacin, which resulted in urethane bond formation in derivative **34**. As in the case of TML-containing conjugates, the Boc-protecting groups were removed with TFA, and the resulting primary amino groups were acylated with cholic acid active ester **22b**, leading to the final conjugate **35**. 

Conjugate **37**, carrying the Lys(Mca) fluorescent probe as a cargo, was prepared analogously as conjugate **35** (Figure 6).

### 2.3. Three Out of Five Umbrella: Cispentacin Conjugates Demonstrate Antifungal In Vitro Activity

The antifungal in vitro activity of conjugates **24a**–**d** and **35** was determined by the serial two-fold dilution method, using the 96-well microtiter plates, in two growth media, RPMI-1640 and YNB-AS, against seven yeast species from the *Candida* genus. The MIC_50_ and MIC_90_ values, i.e., concentrations at which cell growth was inhibited in 50% and 90%, respectively, were measured and compared to those of the intact cispentacin. The concentration values presented in Table 1 and Table 2 are expressed in mM units to allow the comparison of the compounds’ effectiveness at the molecular level. For data expressed in μg mL^−1^ units, such comparison is not possible, since the molecular masses of conjugates (≈1755–2114 D range) are 14–16-fold higher than that of cispentacin (128.3 D).

Data presented in Table 1 indicate that conjugate **24c** demonstrated a growth inhibitory effect in RPMI-1640 medium against all *Candida* spp. Tested, except for *C. krusei*. The MIC_50_ and MIC_90_ values of this compound were comparable to those of cispentacin, although *C. glabrata* was slightly more susceptible to **24c** than to cispentacin, and the same was found for some *C. albicans* clinical isolates. Compounds **24a**, **24b**, **24d**, and **35** were devoid of antifungal activity in this medium.

As shown in Table 2, in minimal YNB-AS medium, a growth inhibitory effect was found for **24c** against *C. albicans* and *C. glabrata*, but some activity was also observed for **24a** and **24b** against *C. albicans*. 

The MIC_50_ and MIC_90_ values of cispentacin in minimal YNB-AS medium were two to three times lower than those in RPMI-1640. The latter medium contains a mixture of proteinogenic amino acids, including L-proline. This amino acid competes with cispentacin for proline permease, thus lowering the growth inhibitory activity of this antifungal. In the case of **24c** conjugate, the difference between its MIC_50_ and MIC_90_ values measured in YNB-AS and in RPMI-1640 was much smaller, if any. On the other hand, the **24a** and **24b** conjugates were inactive in RPMI-1640 but demonstrated some activity in YNB-AS. It is not clear which components of the RPMI-1640, not present in YNB-AS, affected the antifungal activity of these two conjugates. 

A comparison of the growth inhibitory activities of cispentacin and its conjugates with the diwalled molecular umbrella against clinical FLC (fluconazole)-resistant and FLC-susceptible *C. albicans* strains revealed no substantial difference for **24c** and slightly lower activity of cispentacin against FLC-resistant B4 and Gu5 than against their respective FLC-susceptible counterparts B3 and Gu4. This result suggests than cispentacin could be a substrate for drug efflux pumps, Cdr1p/Cdr2p overexpressed in Gu5 and for Mdr1p present in B4, whereas the conjugates, especially **24c**, are not. 

No antifungal activity in both growth media was found for conjugates of molecular umbrella with fluorescent probes, **26a**,**b**, **27**, and **37**.

### 2.4. Molecular Umbrella: Cispentacin Conjugates Are Not Hemolytic

Cispentacin, all five molecular umbrella/cispentacin conjugates **24a**–**d** and **35**, as well as four conjugates with fluorescent probes **26a**,**b**, **27**, and **37** were tested for hemolytic activity against human erythrocytes. Conjugates **24a**–**d**, **26a**,**b**, **27**, **35**, and **37** were tested in the 2000–16 μg mL^−1^ range (2000 μg mL^−1^ corresponds to approximately 1 mM concentration) and cispentacin in the 200–1.6 μg mL^−1^ range. Not more than 7.5% hemolysis at the higher concentration of each compound was found, which means that all conjugates tested are not hemolytic. 

### 2.5. Molecular Umbrella: TML:Nap-NH_2_ Conjugate ***27*** Is Accumulated in C. albicans and C. glabrata but Not in C. krusei Cells

Four conjugates of a molecular umbrella with fluorescent probes, structural analogs of compounds **24a**–**c** and **35**, were constructed in order to use them for studies on the internalization of molecular umbrella conjugates in *Candida* cells. Three of them, namely **26a**, **26b**, and **37**, contained Mca linked through L-lysine or 2,4-diaminobutanoic acid to the TML system and in **27**, the fluorescent probe was Nap-NH_2_, which was connected directly to TML.

For uptake studies, *Candida* cells were treated with a molecular umbrella/fluorescent probe conjugate, and intracellular probe accumulation was monitored by fluorescence microscopy, with λ_exc_ = 438 nm for Nap-NH_2_ and λ_exc_ = 350 nm for Lys(Mca). 

The results of microscopic examination of **27** uptake by *Candida* cells are shown in Figure 2. This conjugate was effectively accumulated in a time-dependent fashion in *C. albicans* and *C. glabrata* cells but not in *C. krusei*. No fluorescent probe accumulation was observed with conjugates **26a**, **26b**, and **37** (images not shown). 

### 2.6. Conjugates Containing TML Are Cleaved in the Model System and in Cell-Free Extract

Conjugates **24a**–**d**, containing the TML system, were tested for susceptibility to enzymatic cleavage in the model system with pig liver esterase and in cell-free extract prepared from *C. albicans* cells. Conjugate **35** containing the *o*-dithiobenzylcarbamoyl linker was tested in the model system with glutathione and in *C. albicans* cell-free extract.

Samples collected from incubation mixtures at time intervals were subjected to HPLC-MS analysis. Results of these analyses of mixtures containing products of digestion of conjugates **24c** and **24d** with pig esterase are shown in Figure 3A,B. In both cases, a peak with time-dependent increasing intensity appeared at a location corresponding to that of the intact cispentacin. Presence of the released cispentacin was confirmed by MS spectrum, where a signal at *m*/*z* = 128.3 is clearly shown. The respective peak was not present in chromatograms of mixtures resulting from pig esterase treatment of **26a** and **26b** but was found in the chromatogram of post-reaction mixture resulting from glutathione action on **35** (not shown).

The analogous HPLC-MS analyses were performed for samples collected at time intervals from mixtures consisting of conjugates **24a**–**d** or **35** dissolved in the cell-free extract of *C. albicans* cells. Results of analyses of mixtures containing conjugates **24c** and **24d** are shown in Figure 4A,B.

Peaks with time-dependent increasing intensity, at locations roughly corresponding to those of the intact cispentacin, are present in both chromatograms. Surprisingly, in the MS spectra, there are not signals with *m*/*z* = 128.3. Instead, the strong signals at *m*/*z* = 145.1 or 145.2 are present. Most likely, these signals may be attributed to any cispentacin derivative formed in the cell-free extract. In the control experiment, cispentacin was incubated in the *C. albicans* cell-free extract, but the MS analysis did not reveal the formation of any compound with *m*/*z* = 145.1 or 145.2.

Neither cispentacin nor the *m*/*z* 145.1/145.2 compound was found in the reaction mixtures containing conjugates **24a**, **24b**, or **35**. It may suggest that these conjugates are not cleaved in *C. albicans* cell-free extract.

## 3. Discussion

Among the five rationally designed conjugates of a diwalled molecular umbrella with cispentacin, joined through the cleavable linker, only one, namely the TML–pimelate system containing **24c**, exhibited antifungal in vitro activity in RPMI-1640 medium under CLSI-recommended conditions. Two analogs of this conjugate, containing the much shorter TML–succinate linker, **24a** and **24b**, appeared active only in minimal YNB-AS medium, while conjugates **24d** and **35** (containing the *o*-dithiobenzylcarbamoyl linker) were inactive in both media. Lack of the antifungal activity of **35** in RPMI-1640 medium could be at least in part explained by the presence of reduced glutathione at 1 mg L^−1^ in this medium; however, YNB-AS does not contain any thiol compounds. Rather unexpectedly, conjugate **35** was not cleaved in the cytoplasmic extract of *C. albicans*, although cleavage was observed in the model system with 5 mM glutathione. No cleavage in cell-free extract was also found for TML-containing conjugates **24a** and **24b**. However, these conjugates were also found resistant to enzymatic cleavage by pig esterase in the model system. Most probably, the succinic acid link between spermidine scaffold and TML appeared to be short, and due to the close vicinity of the molecular umbrella–spermidine scaffold and TML, enzyme access to the ester bond in TML system is hindered. It seems that pimelic acid could be the optimal dicarboxylic linker joining the TML system with the spermidine scaffold. Conjugate **26d**, containing suberic acid, which is only one methylene unit longer than pimelic acid, was not active as an antifungal, which was apparently due to the problems with internalization, since it was found cleavable in the cell-free extract. Notably, pimelate was previously found as the optimal linker in non-cleavable molecular umbrella-Amphotericin B conjugates [10].

Among the three *Candida* species tested, *C. krusei* appeared resistant to all conjugates, including **24c**. Since conjugate **27**, the fluorescent analogue of **24c**, was not accumulated by *C. krusei* cells, one may suspect that **24c** is also not taken up by these cells. We have no explanation for this phenomenon; nevertheless, it shows that the molecular umbrella may not by an universal molecular nanocarrier.

From the results of HPLC-MS analysis of metabolism of conjugates **24c** and **24d** in *C. albicans* cell-free extract, it is clear that the product of a possible cleavage, with RT similar to that of cispentacin but with 17 D higher MW, was formed, while such a product was not observed when intact cispentacin was incubated in the *C. albicans* cell-free extract. In this context, it is worth mentioning that there have not been any previous reports on the possible modification of cispentacin in the cytosol of fungal cells. On the other hand, it was found that the close structural analogue of cispentacin, BAY 10-8888, known also as Icofungipen or PLD-118, was not metabolized by liver microsomes under in vitro conditions [20]. Undoubtedly, further studies are necessary to allow a correct interpretation of this finding, and any speculation on the identity of a possible cispentacin metabolite makes no sense.

So far, molecular umbrellas have been tested as possible molecular nanocarriers exclusively in liposomal or mammalian cell systems [8,21] with no attempts concerning microbial, cell wall-containing cells. The presence of the cell wall surrounding the cytoplasmic membrane might theoretically constitute a physical barrier for molecules as large as molecular umbrellas and their conjugates. In this respect, it is worth mentioning that small but not large dendrimers were used as nanocarriers for some antimicrobials [22] and, on the other hand, tetra- and octawalled molecular umbrellas were not effective as components of conjugates with Amphotericin B [23]. Now, we have been able to demonstrate that at least conjugates incorporating a diwalled molecular umbrella are able to transgress this obstacle in *Candida* cells. This finding opens the possibility of further attempts aimed at the construction of other conjugates more active than those described in this work, possibly with cargo molecules other than cispentacin.

## 4. Materials and Methods

### 4.1. Chemistry

#### 4.1.1. General

All solvents and reagents were used as obtained from commercial sources. ^1^H NMR and ^13^C NMR spectra were obtained at 500 MHz Varian Unity Plus spectrometer, Varian Medical Systems, Darmstadt, Germany. and the deuterated solvents were used as internal locks. FTIR spectra were obtained with a Thermo Electron Nicolet 8700 spectrometer, Thermo Fisher Scientific, Waltham, MA, USA. High-resolution mass spectra were obtained using the Aqilent Technologies 6540 UHD Accurate—Mass Q-TOF LC/MS mass spectrometer, Waldbronn, Germany. Melting points were determined on a melting point apparatus equipped with a thermometer and were uncorrected. Column chromatography was performed with silica gel (0.040–0.063 mm) by using the indicated solvent systems. The following abbreviations are used in reporting NMR data: s—singlet, brs—broad singlet, d—doublet, dd—doublet of doublets, t—triplet, q—quartet, qt—quintet. 

#### 4.1.2. Syntheses

*Dicarboxylic Acid Anhydrides—General Procedure* (**1b**,**c**), First, 19 mmol of a dicarboxylic acid was dissolved in 50 mL of acetic anhydride and refluxed for 4 h. The solvents were removed under reduced pressure, obtaining 18 mmol of a cyclic anhydride.

*Pimelic Anhydride* **1b**, Starting from 3 g (19 mmol) of pimelic acid, 2.64 g (18 mmol, 95%) of pimelic anhydride **1b** was obtained as a light-beige, low-melting solid. ^1^H NMR (500 MHz, DMSO-*d*_6_) δ: 2.50 (t, *J* = 7.3 Hz, 4H), 1.55 (qt, *J* = 7.4 Hz, 4H), 1.33 (qt, *J* = 6.9 Hz, 2H). ^13^C NMR (125 MHz, DMSO-*d*_6_) δ: 170.08, 34.83, 27.85, 23.88.

*Suberic Anhydride* **1c**, Starting from 10 g (57 mmol) of suberic acid, 9 g (56 mmol, 98%) of suberic anhydride **1c** was obtained as a light-beige, low-melting solid. ^1^H NMR (500 MHz, DMSO-*d*_6_) δ: 2.49 (t, *J* = 7.1 Hz, 4H), 1.56 (m, 4H), 1.32 (m, 4H). ^13^C NMR (125 MHz, DMSO-*d*_6_) δ: 169.66, 34.47, 28.22, 24.03.

*Mono-tert-butyl esters* (**3a**–**c**)—*General Procedure*, First, 20 mmol of carboxylic acid anhydride, 2 mmol of DMAP, and 5.88 mmol of NHS were suspended in 75 mL of toluene. Subsequently, 40 mmol of *tert*-butanol and 6 mmol of TEA were added, and the resulting mixture was refluxed for 24 h in an oil bath. The mixture was cooled to room temperature; then, 75 mL of AcOEt was added. The resulting solution was washed with 1M HCl_(aq)_ (3 × 70 mL) and with brine (3 × 70 mL), respectively. The organic layer was dried over anhydrous MgSO_4_, the desiccant was filtered off, and the filtrate was concentrated under reduced pressure. The residue was purified by liquid column chromatography or crystallization.

*Mono-tert-butyl succinate* **3a**, Starting from 25.37 g (0.25 mol) of succinic anhydride **1a**, 21.1 g (0.12 mol, 48%) of monoester **3a** was obtained as a white solid with m.p. 42–44 °C (AcOEt/hexanes). ^1^H NMR (500 MHz, DMSO-*d*_6_) δ: 12.13 (brs, 1H), 2.38 (s, 4H), 1.39 (s, 9H). ^13^C NMR (125 MHz, DMSO-*d*_6_) δ: 173.67, 171.88, 80.08, 30.24, 29.10, 27.31.

*Mono-tert-butyl pimelate* **3b**, Starting from 6 g (42.08 mmol) of pimelic anhydride **1b**, 5.25 g (24.27 mmol, 58%) of monoester **3b** was obtained as a light-yellow oil with R_f_ 0.67 (hexanes/AcOEt, 4/6, *v*/*v*). ^1^H NMR (500 MHz, DMSO-*d*_6_) δ: 11.98 (brs, 1H), 2.18 (m, 4H), 1.50 (m, 4H), 1.40 (s, 9H), 1.27 (m, 2H). ^13^C NMR (125 MHz, DMSO-*d*_6_) δ: 174.86, 172.79, 79.97, 35.08, 34.05, 28.42, 28.22, 24.80, 24.64.

*Mono-tert-butyl suberate* **3c**, Starting from 9 g (58 mmol) of suberic anhydride **1c**, 3.23 g (14.03 mmol, 24%) of monoester **3c** was obtained as colorless oil with R_f_ 0.63 (hexanes/AcOEt/AcOH, 80/20/1, *v*/*v*/*v*). ^1^H NMR (500 MHz, DMSO-*d*_6_) δ: 11.97 (brs, 1H), 2.17 (m, 4H), 1.47 (m, 4H), 1.39 (s, 9H), 1.25 (m, 4H). ^13^C NMR (125 MHz, DMSO-*d*_6_) δ: 174.96, 172.60, 79.63, 35.14, 34.00, 28.64, 28.54, 28.20, 24.90, 24.78.

*4,4,5,7-tetramethylchroman-2-one* **6**, To a mixture of 22.21 g (0.18 mol) of 3,5-dimethylphenol **5** dissolved in 40 mL of methanesulfonic acid, 23 g (0.20 mol) of methyl 3-methylbut-2-enoate **4** was added in one portion. The reaction mixture was stirred and heated at a 70 °C in an oil bath for 18 h. The mixture was cooled to room temperature, poured into 400 mL of cold water, and extracted with ethyl acetate (3 × 150 mL). The organic layer was washed with water (3 × 150 mL), saturated NaHCO_3(aq)_ (3 × 150 mL), and water (2 × 150 mL), respectively. The acetate layer was dried over anhydrous MgSO_4_, the desiccant was removed, and the filtrate was concentrated under reduced pressure. The residue was recrystallized from diethyl ether obtaining 33.12 g (0.16 mol, 90%) of product **6** as colorless crystals, with m.p. 88–91 °C and R_f_ 0.35 (hexanes/AcOEt, 9/1, *v*/*v*). ^1^H NMR (500 MHz, DMSO-*d*_6_) δ: 6.78 (s, 1H), 6.72 (s, 1H), 2.65 (s, 2H), 2.42 (s, 3H), 2.21 (s, 3H), 1.34 (s, 6H). ^13^C NMR (125 MHz, DMSO-*d*_6_) δ: 168.35, 151.79, 137.28, 136.29, 129.46, 127.34, 115.75, 45.00, 35.06, 27.57, 23.00, 20.35. FTIR ν (cm^−1^): 3050, 2900, 1775, 1625, 1575. 

*2-(3-hydroxy-1,1-dimethylpropyl)-3,5-dimethylphenol* **7**, To a stirred suspension of 4 g (0.104 mol) of LiAlH_4_ in 100 mL of dry THF, 5.3 g (26 mmol) of lactone **6** was added in small portions. Once the addition was completed, the reaction mixture was stirred at room temperature for 3 h. The excess of unreacted LiAlH_4_ was decomposed by adding AcOEt, MeOH, and water, respectively. The slurry was filtered off under reduced pressure, and the filtrate was concentrated under reduced pressure. The residue was dissolved in 100 mL of dichloromethane and dried over anhydrous MgSO_4_. After the removal of desiccant and concentration of filtrate under reduced pressure, the residue was recrystallized from dichloromethane. Obtained 3.47 g (17 mmol, 65%) of product **7** as colorless crystals, with m.p. 112–114 °C and R_f_ 0.27 (hexanes/AcOEt, 7/3, *v*/*v*). ^1^H NMR (500 MHz, DMSO-*d*_6_) δ: 8.96 (s, 1H), 6.43 (s, 1H), 6.28 (s, 1H), 4.11 (t, *J* = 4.9 Hz, 1H), 3.20 (m, 2H), 2.35 (s, 3H), 2.07 (s, 3H), 2.04 (t, *J* = 7.9 Hz, 2H), 1.44 (s, 6H). ^13^C NMR (125 MHz, DMSO-*d*_6_) δ: 157.42, 137.25, 135.16, 128.95, 125.79, 116.22, 59.59, 45.23, 40.00, 32.34, 25.98, 20.58.

*2-{3-[(O-tert-butyldimethylsilyl)hydroxy]-1,1-dimethylpropyl}-3,5-dimethylphenol* **8**, The reaction was carried out under argon atmosphere. To an ice-cold stirred mixture of 250 mg (1.2 mmol) of phenol **7** and 244 mg (2 mmol) of DMAP, dissolved in 10 mL of dry dichloromethane, 217 mg (1.44 mmol) of TBDMSCl in 10 mL of dry dichloromethane was added dropwise. The mixture was stirred in an ice bath for 2 h and then for 3 h at room temperature. Subsequently, the mixture was washed with water (2 × 10 mL), 5% solution of NaHCO_3(aq)_ (3 × 10 mL), and water (2 × 10 mL), respectively. The organic layer was dried over anhydrous MgSO_4_, the desiccant was filtered off, and the filtrate was concentrated under reduced pressure. The residue was recrystallized from dichloromethane, obtaining 400 mg (1.2 mmol, 100%) of product **8** as white crystals, with m.p. 106–109 °C and R_f_ 0.52 (hexanes/AcOEt, 9/1, *v*/*v*). ^1^H NMR (500 MHz, DMSO-*d*_6_) δ: 8.99 (s, 1H), 6.43 (s, 1H), 6.28 (s, 1H), 3.39 (t, *J* = 7.4 Hz, 2H), 2.36 (s, 3H), 2.09 (m, 2H), 2.05 (s, 3H), 1.42 (s, 6H), 0.79 (s, 9H), −0.08 (s, 6H). ^13^C NMR (125 MHz, DMSO-*d*_6_) δ: 157.25, 137.27, 134.91, 128.29, 125.93, 115.85, 61.50, 45.50, 32.37, 26.44, 25.86, 20.68, 18.15, −4.71.

*2-{3-[(O-tert-butyldimethylsilyl)hydroxy]-1,1-dimethylpropyl}-O′-acyloyl-3,5-dimethylphenol* (**9a**–**c**)*—General Procedure*, A stirred mixture of 15.5 mmol of phenol, 23.5 mmol of carboxylic acid, and 2.05 mmol of DMAP in 100 mL of dry dichloromethane was cooled in an ice bath to 0 °C, and then, 30.5 mmol of DCC dissolved in 20 mL of dry dichloromethane was added dropwise. The mixture was stirred at room temperature for 48 h. The precipitate of DCU was filtered off, and the filtrate was concentrated under reduced pressure. The residue was dissolved in 100 mL of chloroform and washed with saturated NaHCO_3(aq)_ (2 × 50 mL), 5% NaHSO_4(aq)_ (2 × 50 mL), and water (2 × 50 mL), respectively. The organic layer was dried over anhydrous MgSO_4_, the desiccant was filtered off, and the filtrate was concentrated under reduced pressure. The residue was purified by liquid column chromatography, using a mixture of solvents hexanes/AcOEt, 9/1, *v*/*v* as a mobile phase.

*tert-Butyl-2-{3-[(O-tert-butyldimethylsilyl)hydroxy]-1,1-dimethylpropyl}-3,5-dimethylphenyl succinate* **9a**, Starting from 810 mg (4.65 mmol) of carboxylic acid **3a** and 1 g (3.1 mmol) of phenol **8**, 1.41 g (2.94 mmol, 95%) of ester **9a** was obtained as a colorless oil. R_f_ 0.52 (hexanes/AcOEt, 9/1, *v*/*v*). HRMS-ESI: *m*/*z* calcd. for C_27_H_46_O_5_Si 478.3115; found 479.3251 [M + 1]^+^. ^1^H NMR (500 MHz, DMSO-*d*_6_) δ: 6.82 (s, 1H), 6.55 (s, 1H), 3.42 (t, *J* = 7.2 Hz, 2H), 2.75 (t, *J* = 6.7 Hz, 2H), 2.56 (t, *J* = 6.3 Hz, 2H), 2.48 (s, 3H), 2.17 (s, 2H), 1.96 (t, *J* = 7.2 Hz, 2H), 1.41 (s, 6H), 1.39 (s, 9H), 0.81 (s, 9H), −0.06 (s, 6H). ^13^C NMR (125 MHz, DMSO-*d*_6_) δ: 171.74, 171.44, 150.06, 138.24, 135.58, 134.00, 132.25, 123.37, 80.45, 60.83, 46.02, 39.17, 32.05, 30.07, 28.15, 26.23, 25.15, 20.01, 18.08, −4.87.

*tert-Butyl-2-{3-[(O-tert-butyldimethylsilyl)hydroxy]-1,1-dimethylpropyl}-3,5-dimethylphenyl pimelate* **9b**, Starting from 2.67 g (12 mmol) of carboxylic acid **3b** and 2.58 g (8 mmol) of phenol **8**, 3.40 g (6.52 mmol, 82%) of ester **9b** was obtained as a colorless oil. R_f_ 0.50 (hexanes/AcOEt, 9/1, *v*/*v*). HRMS-ESI: *m*/*z* calcd. for C_30_H_52_O_5_Si 520.3584; found 521.3579 [M + 1]^+^. ^1^H NMR (500 MHz, DMSO-*d*_6_) δ: 6.83 (s 1H), 6.58 (s, 1H), 3.42 (t, *J* = 7.4 Hz, 2H), 2.54 (m, 2H), 2.48 (s, 3H), 2.21 (m, 2H), 2.18 (s, 3H), 1.95 (t, *J* = 7.5 Hz, 2H), 1.63 (qt, *J* = 7.5 Hz, 2H), 1.54 (qt, *J* = 7.4 Hz, 2H), 1.40 (m, 17H), 0.81 (s, 9H), −0.06 (s, 6H). ^13^C NMR (125 MHz, DMSO-*d*_6_) δ: 172.57, 172.44, 150.09, 138.19, 135.77, 134.12, 132.18, 123.59, 79.86, 79.69, 60.63, 45.94, 39.15, 35.03, 34.49, 32.05, 28.31, 28.21, 26.23, 25.30, 24.76, 24.26, 20.12, 18.29.

*tert-Butyl-2-{3-[(O-tert-butyldimethylsilyl)hydroxy]-1,1-dimethylpropyl}-3,5-dimethylphenyl suberate* **9c**, Starting from 3.09 g (13.40 mmol) of carboxylic acid **3c** and 3.60 g (11.16 mmol) of phenol **8**, 4.82 g (9.01 mmol, 81%) of ester **9c** was obtained as a colorless oil. R_f_ 0.64 (hexanes/AcOEt, 9/1, *v*/*v*). HRMS-ESI: *m*/*z* calcd. for C_31_H_54_O_5_Si 534.3741; found 535.3743 [M + 1]^+^. ^1^H NMR (500 MHz, DMSO-*d*_6_) δ: 6.81 (s, 1H), 6.56 (s, 1H), 3.40 (t, *J* = 7.3 Hz, 2H), 2.52 (m, 2H), 2.46 (s, 3H), 2.17 (s, 3H), 1.93 (t, *J* = 7.3 Hz, 2H), 1.62 (qt, *J* = 7.2 Hz, 2H), 1.50 (qt, *J* = 7.5 Hz, 2H), 1.39 (m, 15H), 1.35–1.09 (m, 6H), 0.78 (s, 9H), −0.06 (s, 6H). ^13^C NMR (125 MHz, DMSO-*d*_6_) δ: 172.63, 172.50, 150.04, 138.15, 135.76, 134.07, 132.21, 123.64, 79.71, 60.55, 45.81, 39.15, 35.13, 34.45, 32.01, 28.52, 28.20, 27.80, 26.21, 25.80, 25.32, 24.79, 24.35, 20.13, 18.18, −5.02.

*2-(3-hydroxy-1,1-dimethylpropyl}-O′-acyloyl-3,5-dimethylphenol* (**10a**–**c**)—*General Procedure*, The silyl ether was dissolved in 200 mL of a mixture of THF/H_2_O/AcOH, 1/1/3, *v*/*v*/*v* and stirred at room temperature for 3 h. The solvent was evaporated under reduced pressure. The residue was dissolved in 100 mL of AcOEt and washed with water (30 mL), saturated NaHCO_3(aq)_ (3 × 30 mL), and water (3 × 30 mL), respectively. The organic layer was dried over anhydrous MgSO_4_, the desiccant was filtered off, and the filtrate was concentrated under reduced pressure. The residue was purified by liquid column chromatography, using a mixture of solvents hexanes/AcOEt, 7/3, *v*/*v* as a mobile phase.

*tert-Butyl-2-(3-hydroxy-1,1-dimethylpropyl}-3,5-dimethylphenyl succinate* **10a**, Starting from 204 mg (0.43 mmol) of ester **9a**, 112 mg (0.31 mmol, 72%) of alcohol **10a** was obtained as a colorless oil. R_f_ 0.41 (hexanes/AcOEt, 7/3, *v*/*v*). HRMS-ESI: *m*/*z* calcd. for C_21_H_32_O_5_ 364.2251; found 365.2580 [M + 1]^+^. ^1^H NMR (500 MHz, DMSO-*d*_6_) δ: 6.81 (s, 1H), 6.54 (s, 1H), 4.20 (t, *J* = 5 Hz, 1H), 3.20 (m, 2H), 2.76 (t, *J* = 6.6 Hz, 2H), 2.55 (t, *J* = 6.3 Hz, 2H), 2.48 (s, 3H), 2.15 (s, 3H), 1.90 (t, *J* = 7.7 Hz, 2H), 1.40 (s, 9H), 1.39 (s, 6H). ^13^C NMR (125 MHz, DMSO-*d*_6_) δ: 171.25, 170.92, 149.28, 137.73, 135.04, 133.79, 131.55, 122.79, 79.78, 58.06, 45.43, 38.41, 31.30, 29.68, 27.48, 24.58, 19.37.

*tert-Butyl-2-(3-hydroxy-1,1-dimethylpropyl}-3,5-dimethylphenyl pimelate* **10b**, Starting from 3.40 g (6.52 mmol) of ester **9b**, 1.40 g (3.44 mmol, 53%) of alcohol **10b** was obtained as a colorless oil. R_f_ 0.46 (hexanes/AcOEt, 7/3, *v*/*v*). HRMS-ESI: *m*/*z* calcd. for C_24_H_38_O_5_ 406.2790; found 407.2789 [M + 1]^+^. ^1^H NMR (500 MHz, CDCl_3_) δ: 6.81 (s, 1H), 6.52 (s, 1H), 3.53 (t, *J* = 6.9 Hz, 2H), 2.55 (t, *J* = 7.4 Hz, 2H), 2.52 (s, 3H), 2.24 (t, *J* = 7.3 Hz, 2H), 2.22 (s, 3H), 2.04 (t, *J* = 7.4 Hz, 2H), 1.77 (qt, *J* = 7.5 Hz, 2H), 1.64 (qt, *J* = 7.7 Hz, 2H), 1.48 (s, 6H), 1.46–1.38 (m, 11H). ^13^C NMR (125 MHz, CDCl_3_) δ: 168.41, 168.27, 145.02, 133.71, 131.42, 129.05, 127.73, 118.52, 75.40, 55.76, 41.01, 34.37, 30.58, 30.09, 27.27, 23.86, 23.37, 20.58, 19.99, 19.60, 15.42.

*tert-Butyl-2-(3-hydroxy-1,1-dimethylpropyl}-3,5-dimethylphenyl suberate* **10c**, Starting from 4.82 g (9.01 mmol) of ester **9c**, 3.25 g (7.72 mmol, 86%) of alcohol **10c** was obtained as a colorless oil. R_f_ 0.50 (hexanes/AcOEt, 7/3, *v*/*v*). HRMS-ESI: *m*/*z* calcd. for C_25_H_40_O_5_ 420.2876; found 421.2875 [M + 1]^+^. ^1^H NMR (500 MHz, CDCl_3_) δ: 6.84 (s, 1H), 6.54 (s, 1H), 3.56 (t, *J* = 7.3 Hz, 2H), 2.56 (m, 5H), 2.22 (m, 6H), 2.07 (t, *J* = 7.1 Hz, 2H), 1.78 (qt, *J* = 7.6 Hz, 2H), 1.63 (qt, *J* = 7.4 Hz, 2H), 1.49 (s, 6H), 1.45 (s, 9H), 1.41 (m, 4H). ^13^C NMR (125 MHz, CDCl_3_) δ: 174.14, 173.67, 155.42, 137.84, 135.95, 128.08, 126.93, 116.11, 80.39, 633.25, 40.81, 39.66, 35.51, 34.336, 31.64, 28.67, 28.61, 28.12, 25.60, 24.87, 24.733, 20.24.

*2-(3-oxo-1,1-dimethylpropyl}-O-acyloyl-3,5-dimethylphenol—General Procedure* (**11a**–**c**), The reaction was carried out under argon atmosphere. To a stirred solution of 8.55 mmol of alcohol dissolved in 100 mL of dry dichloromethane, 17.2 mmol of PCC was added in one portion. The reaction mixture was stirred at room temperature for 4 h and then was filtered through silica gel using a mixture of hexane/AcOEt, 7/3, *v*/*v* as eluent.

*tert-Butyl-2-(3-hydroxy-1,1-dimethylpropyl}-3,5-dimethylphenyl succinate* **11a**, Starting from 429 mg (1.18 mmol) of alcohol **10a**, 362 mg (1 mmol, 85%) of aldehyde **11a** was obtained as a light-yellow oil. R_f_ 0.33 (hexanes/AcOEt, 9/1, *v*/*v*). HRMS-ESI: *m*/*z* calcd. for C_21_H_30_O_5_ 362.2091; found 363.2430 [M + 1]^+^. ^1^H NMR (500 MHz, DMSO-*d*_6_) δ: 9.44 (t, *J* = 2.3 Hz, 1H), 6.85 (s, 1H), 6.58 (s, 1H), 2.88 (d, *J* = 2.1 Hz, 2H), 2.77 (t, *J* = 6.6 Hz, 2H), 2.55 (t, *J* = 6.6 Hz, 2H), 2.48 (s, 3H), 2.17 (s, 3H), 1.47 (s, 6H), 1.38 (s, 9H). ^13^C NMR (125 MHz, DMSO-*d*_6_) δ: 202.74, 171.99, 171.61, 149.60, 138.10, 136.12, 133.51, 132.17, 123.56, 80.59, 56.31, 37.96, 31.40, 30.10, 28.09, 25.19, 19.90.

*tert-Butyl-2-(3-hydroxy-1,1-dimethylpropyl}-3,5-dimethylphenyl pimelate* **11b**, Starting from 1.40 g (3.44 mmol) of alcohol **10b**, 1.35 g (3.34 mmol, 97%) of aldehyde **11b** was obtained as a light-yellow oil. R_f_ 0.32 (hexanes/AcOEt, 9/1, *v*/*v*). HRMS-ESI: *m*/*z* calcd. for C_24_H_36_O_5_ 404.2563; found 405.2559 [M + 1]^+^. ^1^H NMR (500 MHz, DMSO-*d*_6_) δ: 9.45 (t, *J* = 2.4 Hz, 1H), 6.86 (s, 1H), 6.61 (s, 1H), 2.81 (d, *J* = 2.2 Hz, 2H), 2.57 (t, *J* = 7.1 Hz, 2H), 2.50 (s, 3H), 2.21 (t, *J* = 7.1 Hz, 2H), 2.18 (s, 3H), 1.64 (qt, *J* = 7.3 Hz, 2H), 1.54 (qt, *J* = 7.6 Hz, 2H), 1.49 (s, 6H), 1.45–1.30 (m, 11H). ^13^C NMR (125 MHz, DMSO-*d*_6_) δ: 203.08, 172.94, 149.63, 138.21, 136.30, 133.61, 132.56, 123.63, 79.90, 56.66, 38.15, 35.03, 34.45, 31.55, 28.30, 28.23, 25.33, 24.74, 24.27, 20.14.

*tert-Butyl-2-(3-hydroxy-1,1-dimethylpropyl}-3,5-dimethylphenyl suberate* **11c**, Starting from 3.25 g (7.72 mmol) of alcohol **10c**, 2.92 g (6.98 mmol, 90%) of aldehyde **11c** was obtained as a light-yellow oil. R_f_ 0.37 (hexanes/AcOEt, 9/1, *v*/*v*). HRMS-ESI: *m*/*z* calcd. for C_25_H_38_O_5_ 418.2719; found 419.2721 [M + 1]^+^. ^1^H NMR (500 MHz, DMSO-*d*_6_) δ: 9.45 (t, *J* = 2.2 Hz, 1H), 6.84 (s, 1H), 6.58 (s, 1H), 2.80 (d, *J* = 2.7 Hz, 2H), 2.57 (t, *J* = 6.6 Hz, 2H), 2.49 (s, 3H), 2.18 (m, 5H), 1.62 (qt, *J* = 7.2 Hz, 2H), 1.50 (qt, *J* = 7.5 Hz, 2H), 1.47 (s, 6H), 1.39 (s, 9H), 1.33 (m, 4H). ^13^C NMR (125 MHz, DMSO-*d*_6_) δ: 203.06, 172.72, 172.69, 149.66, 137.97, 136.42, 133.40, 132.34, 123.64, 79.86, 56.36, 38.09, 35.11, 34.46, 31.53, 28.51, 28.47, 28.20, 25.32, 24.86, 24.35, 20.14.

*3-[2-(O-acyloyl)hydroxyl-4,6-dimethyl]phenyl-3-methylbutanoic acid* (**12a**–**c**)*—General Procedure*, To a stirred, ice-cold mixture of 2.48 mmol of aldehyde, 100 μL of 50% H_2_O_2_ and 80 mg of NaH_2_PO_4_ in 3 mL of MeCN/H_2_O, 3/1, *v*/*v*, 450 mg of NaClO_2_ in 2 mL of water was added dropwise. The reaction mixture was stirred at 0 °C for 1 h and then warmed to room temperature. The excess of oxidizing reagents was decomposed by a dropwise addition of saturated aqueous solution Na_2_S_2_O_3_. Subsequently, the mixture was acidified with 3M HCl to pH 2 and extracted with AcOEt (3 × 30 mL). The organic layer was washed with water (30 mL), brine (2 × 30 mL), and water (30 mL), respectively. The organic layer was dried over anhydrous MgSO_4_, the desiccant was filtered off, and the filtrate was concentrated under reduced pressure. The product was purified by liquid column chromatography using a mixture of solvents hexanes/AcOEt/AcOH.

*3-{2-[O-(4-tert-butoxysuccinoyl)]hydroxyl-4,6-dimethyl}phenyl-3-methylbutanoic acid***12a**, Starting from 1.46 g (3.68 mmol) of aldehyde **11a**, 1.09 g (2.88 mmol, 78%) of carboxylic acid **12a** was obtained as a light-yellow oil. R_f_ 0.46 (hexanes/AcOEt/AcOH, 70/30/1, *v*/*v*/*v*). HRMS-ESI: *m*/*z* calcd. For C_21_H_30_O_6_ 378.2042; found 379.2370 [M + 1]^+^. ^1^H NMR (500 MHz, DMSO-*d*_6_) δ: 11.79 (s, 1H), 6.80 (s, 1H), 6.55 (s, 1H), 2.78 (t, *J* = 6.7 Hz, 2H), 2.71 (s, 2H), 2.58 (t, *J* = 6 Hz, 2H), 2.47 (s, 3H), 2.17 (s, 3H), 1.48 (s, 6H), 1.41 (s, 9H). ^13^C NMR (125 MHz, DMSO-*d*_6_) δ: 173.10, 171.82, 171.56, 149.74, 138.00, 135.70, 134.25, 132.00, 123.17, 80.51, 48.14, 38.45, 31.26, 30.21, 28.36, 25.26, 20.12.

*3-{2-[O-(4-tert-butoxypimeoyl)]hydroxyl-4,6-dimethyl}phenyl-3-methylbutanoic acid***12b**, Starting from 1.35 g (3.34 mmol) of aldehyde **11b**, 740 mg (1.76 mmol, 53%) of carboxylic acid **12b** was obtained as a colorless oil. R_f_ 0.40 (hexanes/AcOEt/AcOH, 80/20/1, *v*/*v*/*v*). HRMS-ESI: *m*/*z* calcd. for C_24_H_36_O_6_ 420.2512; found 421.2515 [M + 1]^+^. ^1^H NMR (500 MHz, DMSO-*d*_6_) δ: 11.83 (brs, 1H), 6.81 (s, 1H), 6.57 (s, 1H), 2.70 (s, 2H), 2.56 (t, *J* = 7.5 Hz, 2H), 2.50 (s, 3H), 2.21 (t, *J* = 7 Hz, 2H), 2.17 (s, 3H), 1.64 (qt, *J* = 7.4 Hz, 2H), 1.54 (qt, *J* = 7.7 Hz, 2H), 1.47 (s, 6H), 1.43–1.30 (m, 11H). ^13^C NMR (125 MHz, DMSO-*d*_6_) δ: 179.37, 171.69, 167.95, 149.83, 138.21, 135.71, 134.46, 132.17, 123.43, 79.85, 79.63, 47.85, 38.69, 35.07, 34.49, 31.48, 28.31, 28.24, 25.28, 24.76, 24.29, 20.13.

*3-{2-[O-(4-tert-butoxysuberoyl)]hydroxyl-4,6-dimethyl}phenyl-3-methylbutanoic acid* **12c**, Starting from 2.92 g (6.98 mmol) of aldehyde **11c**, 2.41 g (5.55 mmol, 80%) of carboxylic acid **12c** was obtained as a colorless oil. HRMS-ESI: *m*/*z* calcd. for C_25_H_38_O_6_ 434.2668; found 435.2672 [M + 1]^+^. ^1^H NMR (500 MHz, DMSO-*d*_6_) δ: 11.81 (s, 1H), 6.79 (s, 1H), 6.56 (s, 1H), 2.70 (s, 2H), 2.55 (t, *J* = 7.2 Hz, 2H), 2.49 (s, 3H), 2.19 (t, *J* = 7.3 Hz, 2H), 2.17 (s, 3H), 1.63 (qt, *J* = 7.4 Hz, 2H), 1.50 (qt, *J* = 7.7 Hz, 2H), 1.46 (s, 6H), 1.39 (s, 9H), 1.32 (m, 4H). ^13^C NMR (125 MHz, DMSO-*d*_6_) δ: 173.01, 172.73, 172.49, 149.86, 138.10, 135.64, 134.38, 132.09, 123.36, 79.83, 47.78, 38.68, 35.12, 34.50, 31.47, 28.54, 28.49, 28.24, 25.29, 24.87, 24.40, 20.15.

*Benzyl 3-{2-[O-(4-tertbutoxyacyloyl)]hydroxyl-4,6-dimethyl}phenyl-3-methylbutanoate* (**13a–c**)—*General Procedure*, To a mixture of 2.64 mmol of carboxylic acid dissolved in 20 mL of dry DMF, 5.28 mmol of potassium bicarbonate was added, and the resulting suspension was allowed to stir at room temperature for 15 min. Subsequently, 4 mmol of benzyl bromide was added in one portion, and then, the reaction mixture was allowed to stir at 40 °C for 3 h. The mixture was cooled to room temperature, and 30 mL of 5% solution of NaHCO_3(aq)_ was added. The resulting mixture was extracted with ethyl acetate (3 × 50 mL), and then, the organic layer was washed with 5% solution of NaHCO_3(aq)_ (2 × 50 mL) and brine (2 × 50 mL), respectively. The organic layer was dried over anhydrous MgSO_4_, the desiccant was filtered off, and the filtrate was concentrated under reduced pressure. The residue was purified by liquid column chromatography using a mixture of solvents hexanes/AcOEt, 8/2, *v*/*v* as a mobile phase.

*Benzyl 3-{2-[O-(4-tertbutoxysuccinoyl)]hydroxyl-4,6-dimethyl}phenyl-3-methylbutanoate* **13a**, Starting from 1 g (2.64 mmol) of carboxylic acid **12a**, 1 g (2.13 mmol, 81%) of ester **13a** was obtained as a light-yellow oil. R_f_ 0.37 (hexanes/AcOEt, 9/1, *v*/*v*). HRMS-ESI: *m*/*z* calcd. for C_28_H_36_O_6_ 468.2512; found 469.2840 [M + 1]^+^. ^1^H NMR (500 MHz, CDCl_3_) δ: 7.13 (m, 3H), 7.23 (m, 2H), 6.80 (s, 1H), 6.62 (s, 1H), 5.02 (s, 2H), 2.93 (s, 2H), 2.81 (t, *J* = 6.6 Hz, 2H), 2.64 (t, *J* = 7 Hz, 2H), 2.52 (s, 3H), 2.25 (s, 3H), 1.59 (s, 6H), 1.49 (s, 9H). ^13^C NMR (125 MHz, CDCl_3_) δ: 171.61, 171.55, 171.31, 149.50, 139.97, 136.16, 136.02, 133.38, 132.41, 128.39, 128.24, 127.98, 123.00, 80.71, 65.92, 48.04, 38.96, 31.40, 30.15, 27.92, 25.42, 20.29.

*Benzyl 3-{2-[O-(4-tertbutoxypimeoyl)]hydroxyl-4,6-dimethyl}phenyl-3-methylbutanoate* **13b**, Starting from 750 mg (1.78 mmol) of carboxylic acid **12b**, 680 mg (1.33 mmol, 75%) of ester **13b** was obtained as a colorless oil. R_f_ 0.44 (hexanes/AcOEt, 9/1, *v*/*v*). HRMS-ESI: *m*/*z* calcd. for C_31_H_42_O_6_ 510.2981; found 511.2983 [M + 1]^+^. ^1^H NMR (500 MHz, CDCl_3_) δ: 7.32 (m, 3H), 7.21 (m, 2H), 6.80 (s, 1H), 6.58 (s, 1H), 5.02 (s, 2H), 2.53 (m, 5H), 2.26 (m, 5H), 1.76 (qt, *J* = 7.6 Hz, 2H), 1.35 (qt, *J* = 7.7 Hz, 2H), 1.58 (s, 6H), 1.53–1.39 (m, 11H). ^13^C NMR (125 MHz, CDCl_3_) δ: 172.99, 172.44, 171.56, 149.47, 138.00, 136.13, 136.01, 133.38, 132.36, 128.36, 128.24, 127.97, 123.04, 80.06, 66.06, 47.90, 39.05, 35.30, 34.88, 31.57, 28.60, 28.18, 25.28, 24.86, 24.39, 20.22.

*Benzyl 3-{2-[O-(4-tertbutoxysuberoyl)]hydroxyl-4,6-dimethyl}phenyl-3-methylbutanoate* **13c**, Starting from 750 mg (1.78 mmol) of carboxylic acid **12c**, 680 mg (1.33 mmol, 75%) of ester **13c** was obtained as a colorless oil. R_f_ 0.44 (hexanes/AcOEt, 9/1, *v*/*v*). HRMS-ESI: *m*/*z* calcd. for C_32_H_44_O_6_ 524.3138; found 525.3134 [M + 1]^+^. ^1^H NMR (500 MHz, CDCl_3_) δ: 7.32 (m, 3H), 7.21 (m, 2H), 6.80 (s, 1H), 6.57 (s, 1H), 5.01 (s, 2H), 2.89 (s, 2H), 2.51 (m, 5H), 2.25 (s, 3H), 2.23 (t, *J* = 7.6 Hz, 2H), 1.73 (qt, *J* = 7.4 Hz, 2H), 1.61 (qt, *J* = 7.5 Hz, 2H), 1.57 (s, 6H), 1.47 (s, 9H), 1.38 (m, 4H). ^13^C NMR (125 MHz, CDCl_3_) δ: 173.18, 172.60, 171.60, 159.39, 149.50, 138.01, 136.14, 136.01, 133.33, 132.37, 128.37, 128.22, 127.99, 123.03, 80.01, 66.02, 47.78, 38.96, 35.49, 34.95, 31.52, 28.86, 28.76, 28.14, 25.33, 24.90, 24.51, 20.28.

*Benzyl 3-[2-[O-acyloyl]hydroxyl-4,6-dimethyl]phenyl-3-methylbutanoate* (**14a**–**c**)—*General Procedure*, First, 2.13 mmol of *tert*-butyl ester was dissolved in 20 mL of a mixture of DCM/TFA, 3/1, *v*/*v* and allowed to stir at room temperature for 1 h. The solvents were removed under reduced pressure, and the residue was purified by liquid column chromatography, using a mixture of solvents hexanes/AcOEt/AcOH, 70/30/1, *v*/*v*/*v* as a mobile phase.

*Benzyl 3-[2-[O-succinoyl]hydroxyl-4,6-dimethyl]phenyl-3-methylbutanoate* **14a**, Starting from 1 g (2.13 mmol) of ester **13a**, 820 mg (2 mmol, 94%) of carboxylic acid **14a** was obtained as a light-yellow oil. HRMS-ESI: *m*/*z* calcd. for C_24_H_28_O_6_ 412.1892; found 413.1950 [M + 1]^+^. ^1^H NMR (500 MHz, DMSO-*d*_6_) δ: 12.34 (brs, 1H), 7.31 (m, 3H), 7.18 (m, 2H), 6.79 (s, 1H), 6.57 (s, 1H), 4.95 (s, 2H), 2.90 (s, 2H), 2.74 (t, *J* = 6 Hz, 2H), 2.56 (t, *J* = 6.9 Hz, 2H), 2.47 (s, 3H), 2.18 (s, 3H), 1.47 (s, 6H). ^13^C NMR (125 MHz, DMSO-*d*_6_) δ: 173.90, 172.07, 171.42, 150.10, 138.06, 136.81, 135.77, 134.13, 132.30, 128.71, 128.25 123.30, 65.52, 47.73, 38.83, 31.37, 30.33, 28.89, 25.36, 20.19.

*Benzyl 3-[2-[O-pimeoyl]hydroxyl-4,6-dimethyl]phenyl-3-methylbutanoate* **14b**, Starting from 680 mg (1.33 mmol) of ester **13b**, 490 mg (1.08 mmol, 81%) of carboxylic acid **14b** was obtained as a colorless oil. R_f_ 0.42 (hexanes/AcOEt/AcOH, 70/30/1, *v*/*v*/*v*). HRMS-ESI: *m*/*z* calcd. for C_27_H_34_O_6_ 454.2355; found 455.2361 [M + 1]^+^. ^1^H NMR (500 MHz, DMSO-*d*_6_) δ: 12.03 (brs, 1H), 7.31 (m, 3H), 7.18 (m, 2H), 6.80 (s, 1H), 6.58 (s, 1H), 4.95 (s, 2H), 2.86 (s, 2H), 2.52 (m, 2H), 2.46 (s, 3H), 2.21 (t, *J* = 7.5 Hz, 2H), 2.19 (s, 3H), 1.60 (qt, *J* = 7.8 Hz, 2H), 1.52–1.41 (m, 8H), 1.33 (m, 2H). ^13^C NMR (125 MHz, DMSO-*d*_6_) δ: 174.92, 172.48, 171.26, 149.83, 138.02, 136.46, 135.85, 133.85, 132.18, 128.73, 128.35, 123.53, 65.68, 47.70, 39.05, 34.40, 33.97, 31.48, 28.44, 25.33, 24.63, 24.26, 20.17.

*Benzyl 3-[2-[O-suberoyl]hydroxyl-4,6-dimethyl]phenyl-3-methylbutanoate* **14c**, Starting from 2.36 g (4.50 mmol) of ester **13c**, 1.97 g (4.20 mmol, 93%) of carboxylic acid **14c** was obtained as a colorless oil. R_f_ 0.48 (hexanes/AcOEt/AcOH, 70/30/1, *v*/*v*/*v*). HRMS-ESI: *m*/*z* calcd. for C_28_H_36_O_6_ 468.2512; found 469.2516 [M + 1]^+^. ^1^H NMR (500 MHz, DMSO-*d*_6_) δ: 11.97 (brs, 1H), 7.30 (m, 3H), 7.18 (m, 2H), 6.78 (s, 1H), 6.57 (s, 1H), 4.94 (s, 2H), 2.84 (s, 2H), 2.50 (m, 2H), 2.45 (s, 3H), 2.18 (m, 5H), 1.58 (qt, *J* = 6.4 Hz, 2H), 1.47 (m, 8H), 1.29 (m, 4H). ^13^C NMR (125 MHz, DMSO-*d*_6_) δ: 174.91, 172.50, 171.27, 149.78, 138.05, 136.47, 135.88, 133.75, 132.14, 128.68, 128.29, 123.50, 65.59, 47.58, 38.98, 34.48, 34.02, 31.47, 28.62, 28.58, 25.31, 24.76, 24.39, 20.18.

*N*^1^,*N*^7^*-bis(tert-butoxycarbonyl)spermidine* **16**, First, 1.23 g (8.47 mmol) of spermidine **15** was dissolved in 50 mL of dry THF and cooled in an ice bath to 0 °C. Then, 4.17 g (16.94 mmol) of Boc-ON, dissolved in 50 mL of dry THF, was added dropwise. The mixture was allowed to stir at 0 °C for 4 h, and then, solvents were evaporated under reduced pressure. The residue was dissolved in 50 mL of Et_2_O and washed with saturated NaOH_(aq)_ until the yellow color disappeared. The organic layer was dried over anhydrous MgSO_4_, the desiccant was filtered off, and the filtrate was concentrated under reduced pressure. The white crystalline residue was recrystallized from Et_2_O obtaining 1.88 g (5.44 mmol, 64%) of product **16** as white solid, with m.p. 83–85 °C. ^1^H NMR (500 MHz, CDCl_3_) δ: 5.22 (brs, 1H), 4.88 (brs, 1H), 3.21 (m, 2H), 3.14 (m, 2H), 2.67 (t, *J* = 6.6 Hz, 2H), 2.61 (t, *J* = 6.2 Hz, 2H), 1.65 (qt, *J* = 6.5 Hz, 2H), 1.53 (m, 4H), 1.45 (s, 18H). ^13^C NMR (125 MHz, CDCl_3_) δ: 156.09, 156.01, 78.94, 49.45, 47.75, 40.53, 39.29, 29.88, 28.44, 27.84, 27.41.

*Boc*_2_*-spermidine-‘trimethyl lock’ benzyl ester building block* (**18a**–**c**)*—General Procedure*, To the solution of 2.42 mmol of carboxylic acid **14a**–**c** and 3.06 mmol of NHS in 20 mL of dry DCM, 4.34 mmol of DCC dissolved in 5 mL of dry DCM was added dropwise, and then, the mixture was allowed to stir at room temperature for 24 h. The precipitate of DCU was filtered off under reduced pressure, and the filtrate was diluted with 50 mL of DCM. Subsequently, the mixture was washed with water (2 × 30 mL), saturated solution of NaHCO_3(aq)_ (2 × 30 mL), and water (2 × 30 mL), respectively. The organic layer was dried over anhydrous MgSO_4_, the desiccant was filtered off, and the filtrate was concentrated under reduced pressure. The residue was dissolved in 50 mL of dry DCM, and 6.72 mmol of DIPEA and 3.2 mmol of Boc_2_-spermidine 16 were added. The mixture was allowed to stir at room temperature for 5 h; then, 20 mL of DCM was added, and the resulting solution was washed with brine (2 × 30 mL), 5% solution of NaHSO_4(aq)_ (2 × 30 mL), and brine (2 × 30 mL), respectively. The organic layer was dried over anhydrous MgSO_4_, the desiccant was filtered off, and the filtrate was concentrated under reduced pressure. The residue was purified by liquid column chromatography, using a mixture of solvents hexanes/AcOEt, 4/6, *v*/*v* as a mobile phase.

*N*^1^,*N*^7^*-bis(tert-butoxycarbonyl)-N*^3^*-{4-[2-(2-benzyloxycarbonyl-1,1-dimethylethyl)-3,5-dimethyl]phenoxy-1,4-dioxo}butylspermidine* **18a**, Starting from 1 g (2.42 mmol) of carboxylic acid **14a** and 1.1 g (3.2 mmol) of Boc_2_-spermidine **16**, 1.1 g (1.4 mmol, 68%) of product **18a** was obtained as a colorless oil with R_f_ 0.50 (hexanes/AcOEt, 4/6, *v*/*v*). HRMS-ESI: *m*/*z* calcd. for C_41_H_61_N_3_O_9_ 739.4408; found 740.4537 [M + 1]^+^. ^1^H NMR (500 MHz, CD_3_OD) δ: 7.29 (m, 3H), 7.16 (m, 2H), 6.79 (s, 1H), 6.63 (m, 1H), 4.95 (s, 2H), 3.36 (m, 4H), 3.12–2.97 (m, 4H), 2.93 (s, 2H), 2.83 (m, 2H), 2.75 (m, 2H), 2.49 (s, 3H), 2.22 (s, 3H), 1.92–1.36 (m, 30H). ^13^C NMR (125 MHz, CD_3_OD) δ: 172.47, 172.42, 172.13, 171.88, 171.72, 157.11, 156.95, 149.70, 137.80, 136.08, 135.83, 133.14, 131.74, 127.97, 127.82, 127.57, 122.73, 78.09, 65.49, 45.49, 45.23, 43.08, 39.60, 39.35, 38.77, 37.39, 37.19, 33.40, 30.87, 29.91, 28.72, 27.38, 26.87, 25.48, 25.36, 24.66, 24.55, 24.16, 18.95.

*N*^1^,*N*^7^*-bis(tert-butoxycarbonyl)-N*^3^*-{4-[2-(2-benzyloxycarbonyl-1,1-dimethylethyl)-3,5-dimethyl]phenoxy-1,7-dioxo}heptylspermidine* **18b**, Starting from 490 mg (1.08 mmol) of carboxylic acid **14b** and 374 mg (1.08 mmol) of Boc_2_-spermidine **16**, 370 mg (0.47 mmol, 44%) of product **18b** was obtained as a colorless oil with R_f_ 0.47 (hexanes/AcOEt, 4/6, *v*/*v*). HRMS-ESI: *m*/*z* calcd. for C_44_H_67_N_3_O_9_ 781.4877; found 782.4851 [M + 1]^+^. ^1^H NMR (500 MHz, CDCl_3_) δ: 7.30 (m, 3H), 7.19 (m, 2H), 6.78 (s, 1H), 6.57 (s, 1H), 5.00 (s, 2H), 3.46–3.02 (m, 8H), 2.89 (s, 2H), 2.54 (t, *J* = 7.2 Hz, 2H), 2.51 (s, 3H), 2.33 (t, *J* = 7.6 Hz, 2H), 2.24 (s, 3H), 1.83–1.38 (m, 36H). ^13^C NMR (125 MHz, CDCl_3_) δ: 173.14, 172.53, 172.32, 171.62, 156.08, 149.51, 138.02, 136.11, 136.01, 133.36, 132.33, 128.36, 128.21, 127.97, 123.05, 79.34, 78.77, 77.25, 65.98, 47.83, 47.40, 45.51, 42.38, 39.94, 39.06, 37.30, 34.86, 33.98, 32.83, 31.54, 29.92, 28.95, 28.50, 28.43, 27.97, 27.66, 26.15, 25.30, 25.11, 24.96, 24.48, 20.28, 14.21.

*N*^1^,*N*^7^*-bis(tert-butoxycarbonyl)-N*^3^*-{4-[2-(2-benzyloxycarbonyl-1,1-dimethylethyl)-3,5-dimethyl]phenoxy-1,8-dioxo}octylspermidine* **18c**, Starting from 1.97 g (4.20 mmol) of carboxylic acid **14c** and 1.55 g (4.50 mmol) of Boc_2_-spermidine **16**, 2.03 g (2.55 mmol, 61%) of product **18c** was obtained as a colorless oil with R_f_ 0.45 (hexanes/AcOEt, 4/6, *v*/*v*). HRMS-ESI: *m*/*z* calcd. for C_45_H_69_N_3_O_9_ 795.5034; found 796.5126 [M + 1]^+^. ^1^H NMR (500 MHz, CDCl_3_) δ: 7.30 (m, 3H), 7.18 (m, 2H), 6.78 (s, 1H), 6.55 (s, 1H), 5.43 (m, 1H), 4.99 (s, 2H), 4.69 (m, 1H), 3.44–3.01 (m, 8H), 2.88 (s, 2H), 2.50 (m, 5H), 2.30 (m, 2H), 2.24 (s, 3H), 1.77–1.62 (m, 6H), 1.61–1.52 (m, 8H), 1.51–1.31 (m, 24H). ^13^C NMR (125 MHz, CDCl_3_) δ: 167.89, 166.80, 166.45, 151.25, 144.79, 133.27, 131.37, 131.23, 128.59, 127.58, 123.61, 123.45, 123.24, 118.30, 74.13, 61.07, 55.66, 42.98, 42.66, 40.65, 37.51, 35.07, 34.31, 32.36, 30.17, 28.16, 26.72, 24.43, 23.68, 22.91, 21.38, 20.55, 19.73, 16.31, 15.53.

*Boc*_2_*-spermidine-‘trimethyl lock’ carboxylic acid building block* (**19a**–**c**)—*General Procedure*, First, 1.64 mmol of benzyl ester was dissolved in 30 mL of THF, and then, 500 mg of 10% Pd/C was added. The mixture was allowed to stir at room temperature under hydrogen (balloon) atmosphere for 4 h. Subsequently, the catalyst was filtered off under reduced pressure through a thin layer of celite. The filtrate was concentrated under reduced pressure, obtaining 1.41 mmol of carboxylic acid **19**. 

*N*^1^,*N*^7^*-bis(tert-butoxycarbonyl)-N*^3^*-{4-[2-(2-carboxy-1,1-dimethylethyl)-3,5-dimethyl]phenoxy-1,4-dioxo}butylspermidine* **19a**, Starting from 1.21 g (1.64 mmol) of ester **18a**, 918 mg (1.41 mmol, 86%) of carboxylic acid **19a** was obtained as a colorless oil, with R_f_ 0.73 (CHCl_3_/MeOH/H_2_O, 65/10/1, *v*/*v*/*v*). HRMS-ESI: *m*/*z* calcd. for C_34_H_55_N_3_O_9_ 649.3938; found 650.3858 [M + 1]^+^. ^1^H NMR (500 MHz, CD_3_OD) δ: 6.82 (s, 1H), 6.61 (m, 1H), 3.38 (m, 4H), 3.09 (m, 2H), 3.01 (m, 2H), 2.87 (m, 2H), 2.84 (s, 2H), 2.80 (m, 2H), 2.55 (s, 3H), 2.20 (s, 3H), 1.90–1.28 (m, 30H). ^13^C NMR (125 MHz, CD_3_OD) δ: 174.05, 172.56, 172.51, 172.20, 171.80, 149.62, 137.85, 135.69, 133.66, 131.66, 122.66, 78.12, 47.35, 45.39, 45.17, 43.07, 39.55, 39.36, 38.39, 37.38, 37.23, 33.34, 30.64, 29.95, 27.56, 27.40, 26.84, 25.49, 25.33, 24.68, 24.55, 24.18, 18.91.

*N*^1^*,N*^7^*-bis(tert-butoxycarbonyl)-N*^3^*-{4-[2-(2-carboxy-1,1-dimethylethyl)-3,5-dimethyl]phenoxy-1,4-dioxo}heptylspermidine***19b**, Starting from 350 mg (0.448 mmol) of ester **18b**, 298 mg (0.431 mmol, 96%) of carboxylic acid **19b** was obtained as a light-yellow oil. HRMS-ESI: *m*/*z* calcd. for C_37_H_61_N_3_O_9_ 691.4408; found 692.4414 [M + 1]^+^. ^1^H NMR (500 MHz, DMSO-*d*_6_) δ: 11.70 (brs, 1H), 6.80 (s, 1H), 6.58 (s, 1H), 3.21 (m, 4H), 2.91 (m, 4H), 2.70 (s, 2H), 2.56 (t, *J* = 7.1 Hz, 2H), 2.50 (s, 3H), 2.27 (m, 2H), 2.17 (s, 3H), 1.78–1.20 (m, 36H). ^13^C NMR (125 MHz, DMSO-*d*_6_) δ: 173.04, 172.51, 172.04, 171.83, 156.20, 149.87, 138.09, 135.74, 134.31, 132.16, 123.41, 107.65, 79.74, 67.18, 47.87, 38.67, 34.52, 33.80, 32.38, 31.47, 29.36, 28.74, 28.30, 27.58, 27.25, 26.34, 25.29, 25.18, 24.89, 24.56, 24.09, 22.20, 20.17.

*N*^1^*,N*^7^*-bis(tert-butoxycarbonyl)-N*^3^*-{4-[2-(2-carboxy-1,1-dimethylethyl)-3,5-dimethyl]phenoxy-1,4-dioxo}octylspermidine* **19c**, Starting from 2.03 g (2.55 mmol) of ester **18c**, 1.95 g (2.50 mmol, 98%) of carboxylic acid **19c** was obtained as a light-yellow oil with R_f_ 0.64 (CHCl_3_/MeOH/H_2_O, 65/10/1, *v*/*v*/*v*). HRMS-ESI: *m*/*z* calcd. for C_38_H_63_N_3_O_9_ 705.4564; found 706.4572 [M + 1]^+^. ^1^H NMR (500 MHz, DMSO-*d*_6_) δ: 11.84 (brs, 1H), 6.90–6.83 (m, 1H), 6.78–6.73 (m, 2H), 3.20 (m, 4H), 2.90 (m, 4H), 2.69 (s, 2H), 2.56 (t, *J* = 7.4 Hz, 2H), 2.49 (s, 3H), 2.24 (m, 2H), 2.17 (s, 3H), 1.62 (m, 3H), 3.72 (m, 10H), 3.45 (m, 26H). ^13^C NMR (125 MHz, DMSO-*d*_6_) δ: 173.08, 172.50, 172.08, 171.75, 156.03, 149.82, 138.11, 135.62, 134.27, 132.07, 123.44, 79.62, 47.86, 47.33, 44.97, 43.03, 38.70, 38.06, 37.87, 34.57, 32.45, 31.44, 28.88, 28.69, 28.34, 27.54, 26.31, 25.30, 24.49, 20.15.

*Trisodium O,O′-bis(sulfate)deoxycholate* **21a**, To the solution of 1 g (2.55 mmol) of deoxycholic acid **20a** in 10 mL of dry DMF, 7.32 g (46 mmol) of SO_3_/Py complex was added in one portion. The mixture was stirred at room temperature for 24 h. Subsequently, the mixture was alkalinized to pH 8 with a saturated solution of NaHCO_3(aq)_. The solvents were evaporated under reduced pressure, the residue was suspended in 100 mL of MeOH, and then inorganic salts were filtered off. The filtrate was concentrated under reduced pressure, which was followed by treatment with MeCN. The formed precipitate was filtered off, obtaining 1.55 g (2.50 mmol, 98%) of product **21a** as a light-beige solid with m.p. 202–204 °C and R_f_ 0.39 (CHCl_3_/MeOH/H_2_O, 7/4/1, *v*/*v*/*v*). ^1^H NMR (500 MHz, CD_3_OD) δ: 4.86 (s, 1H), 4.26 (m, 1H), 2.36–2.20 (m, 2H), 2.08 (m, 1H), 2.00–1.71 (m, 10H), 1.69–1.54 (m, 2H), 1.54–1.23 (m, 8H), 1.23–0.90 (m, 9H), 0.76 (s, 3H). ^13^C NMR (125 MHz, CD_3_OD) δ: 182.28, 81.21, 79.15, 45.99, 42.35, 36.06, 35.93, 35.60, 35.36, 35.14, 34.00, 33.62, 33.30, 32.56, 30.23, 28.95, 27.34, 27.05, 26.05, 24.68, 23.49, 22.30, 16.95, 11.61.

*Tetrasodium O,O′,O″-tris(sulfate)cholate* **21b**, Compound **21b** was prepared in a similar manner as **21a**. Starting from 1 g (2.45 mmol) of cholic acid **20b** and 10.5 g (66 mmol) of SO_3_/Py complex, 1.49 g (2 mmol, 82%) of product **21b** was obtained as a light-beige solid. ^1^H NMR (500 MHz, CD_3_OD) δ: 4.68 (s, 1H), 4.46 (m, 1H), 4.15 (m, 1H), 2.52–2.21 (m, 5H), 2.18–1.62 (m, 12H), 1.50–1.22 (m, 5H), 1.17–0.90 (m, 8H), 0.78 (s, 3H). ^13^C NMR (125 MHz, CD_3_OD) δ: 182.48, 81.06, 79.52, 76.71, 45.90, 42.19, 42.05, 39.34, 36.19, 35.69, 35.39, 35.10, 34.14, 32.49, 30.30, 29.10, 27.59, 27.44, 27.27, 24.55, 22.52, 21.82, 17.01, 11.61.

*Trisodium O,O′-bis(sulfate)deoxycholate active ester* **22a**, To the solution of 1.5 g (2.42 mmol) of deoxycholic acid derivative **21a** in 30 mL of dry DMF, 355 μL (2.04 mmol) of DIPEA and 916 mg (3.03 mmol) of DEPBT were added, respectively. The mixture was stirred at room temperature for 2 h and then concentrated under reduced pressure. The oily residue was treated with MeCN and the formed precipitate was filtered off, obtaining 1.35 g (1.57 mmol, 65%) of crude active ester **22a** as a light-yellow solid, which was used in the next step without further purification.

*Tetrasodium O,O′,O″-tris(sulfate)cholate active ester* **22b**, Active ester **22b** was prepared in a similar manner as **22a**. Starting from 1.50 g (2.04 mmol) of cholic acid derivative **21b**, 1.16 g (1.35 mmol, 66%) of active ester **22b** was obtained as a light-yellow solid, which was used in the next step without further purification.

*Cholic acid active ester* **22c**, First, 15 g (37 mmol) of cholic acid was dissolved in 250 mL of dry THF, which was followed by the addition of 4.26 g (37 mmol) of NHS. Subsequently, 9.16 g (44 mmol) of DCC dissolved in 30 mL of dry THF was added dropwise. The mixture was stirred at room temperature for 24 h; then, the precipitated DCU was filtered off under reduced pressure, and the filtrate was concentrated under reduced pressure. The residue was dissolved in 300 mL of CHCl_3_ and washed with a saturated solution of NaHCO_3(aq)_ (2 × 200 mL) and water (2 × 200 mL). The organic layer was dried over anhydrous MgSO_4_, the desiccant was filtered off, and the filtrate was concentrated under reduced pressure, obtaining 10.61 g (21 mmol, 57%) of crude active ester **22c**, which was used in the next step without further purification.

*Molecular umbrella–‘trimethyl lock’–cargo conjugates* (**24a**–**d**)—*General Procedure*, To a solution of 2.17 mmol of carboxylic acid **19a**–**c** in 20 mL of dry DMF, 568 μL of DIPEA and 3.26 mmol of DEPBT were added, respectively. The mixture was stirred at room temperature for 2 h. Then, 100 mL of CHCl_3_ was added, and the resulting mixture was washed with 1 M solution of HCl_(aq)_ (2 × 75 mL), brine (2 × 75 mL), saturated solution of NaHCO_3(aq)_ (2 × 75 mL), and brine (2 × 75 mL). The organic layer was dried over anhydrous MgSO_4_, the desiccant was filtered off, and the filtrate was concentrated under reduced pressure. The crude active ester was roughly purified by liquid column chromatography, using a mixture of solvents hexanes/AcOEt, 4/6, *v*/*v* as a mobile phase. Then, 0.70 mmol of active ester was dissolved in 20 mL of dry DMF, and then 1.1 mL of DIPEA and 1.05 mmol of cispentacin were added, respectively. The mixture was stirred at room temperature for 2 h. Subsequently, 100 mL of CHCl_3_ was added, and the resulting solution was washed with 1 M solution of HCl_(aq)_ (3 × 50 mL) and brine (3 × 50 mL). The organic layer was dried over anhydrous MgSO_4_, the desiccant was filtered off, and the filtrate was concentrated under reduced pressure, obtaining crude product **23a**–**c**. Product **23a**–**c** was dissolved in a mixture of DCM/TFA, 3/1, *v*/*v* and allowed to stir at room temperature for 1 h. Then, the solvents were evaporated under reduced pressure, obtaining the crude deprotection product which was used in the next step without further purification. To the solution of 0.30 mmol of deprotected spermidine derivative in dry DMF, 627 μL of DIPEA and 0.60 mmol of active ester **22a** or **22b** were added, respectively. The mixture was stirred at room temperature for 24 h; then, the solvents were evaporated under reduced pressure, and the residue was treated with MeCN. The precipitate was collected by filtration under reduced pressure and purified by liquid column chromatography.

*Molecular umbrella–cispentacin conjugate* **24a**, Starting from 700 mg (1.1 mmol) of carboxylic acid **19a**, 316 mg (0.18 mmol, 27%) of conjugate **24a** was obtained as a white solid, with R_f_ 0.32 (CHCl_3_/MeOH/H_2_O, 7/4/1, *v*/*v*/*v*). ^1^H NMR (500 MHz, CD_3_OD) δ: 6.82 (s, 1H), 6.67 (s, 1H), 4.67 (s, 2H), 4.28 (m, 3H), 2.44 (m, 4H), 3.21 (m, 4H), 2.96 (m, 3H), 2.83 (m, 4H), 2.71 (m, 1H), 2.60–4.42 (m, 4H), 2.36–2.07 (m, 9H), 2.01–0.86 (m, 74H), 0.76 (s, 6H). ^13^C NMR (125 MHz, CD_3_OD) δ: 175.77, 175.53, 173.31, 172.39, 172.03, 163.52, 150.07, 138.06, 136.20, 133.59, 131.90, 122.96, 81.21, 79.37, 52.15, 48.51, 45.91, 45.36, 43.55, 42.31, 39.66, 38.50, 36.73, 36.52, 35.81, 35.15, 33.91, 33.63, 33.30, 32.64, 31.69, 31.26, 30.07, 27.54, 28.28, 27.16, 27.00, 26.67, 26.24, 26.07, 25.66, 24.77, 24.59, 23.51, 22.30, 21.46, 18.89, 16.83, 11.59.

*Molecular umbrella–cispentacin conjugate* **24b**, Starting from 525 mg (0.69 mmol) of carboxylic acid **19a**, 617 mg (0.31 mmol, 45%) of conjugate **24b** was obtained as a white solid, with R_f_ 0.29 (CHCl_3_/MeOH/H_2_O, 7/4/1, *v*/*v*/*v*). ^1^H NMR (500 MHz, CD_3_OD) δ: 6.82 (s, 1H), 6.65 (s, 1H), 4.66 (s, 2H), 4.45 (m, 2H), 4.25 (m, 1H), 4.14 (m, 2H), 3.41 (m, 4H), 3.19 (m, 4H), 2.97–2.65 (m, 6H), 2.53 (m, 3H), 2.48–2.18 (m, 14H), 2.11 (m, 6H), 2.03–1.17 (m, 46H), 1.15–0.86 (m, 16H), 0.76 (s, 6H). ^13^C NMR (125 MHz, CD_3_OD) δ: 176.35, 175.69, 175.55, 175.41, 173.28, 172.15, 171.98, 171.87, 150.00, 138.02, 136.18, 133.57, 131.93, 122.98, 116.89, 80.77, 79.37, 76.43, 52.09, 48.48, 46.12, 45.84, 45.25, 43.42, 42.23, 41.89, 39.47, 39.16, 38.54, 38.44, 36.47, 35.67, 35.40, 35.10, 34.10, 32.81, 32.62, 31.72, 31.28, 31.12, 30.29, 30.06, 29.01, 28.18, 27.58, 27.42, 27.30, 27.16, 27.00, 26.29, 25.64, 24.59, 24.46, 22.49, 21.78, 21.48, 18.93, 16.88, 11.58.

*Molecular umbrella–cispentacin conjugate* **24c**, Starting from 270 mg (0.39 mmol) of carboxylic acid **19b**, 187 mg (0.09 mmol, 23%) of conjugate **24c** was obtained as a white solid with R_f_ 0.25 (CHCl_3_/MeOH/H_2_O, 7/4/1, *v*/*v*/*v*). ^1^H NMR (500 MHz, CD_3_OD) δ: 6.84(s, 1H), 6.61 (s, 1H), 4.68 (s, 2H), 4.47 (s, 2H), 4.26 (m, 1H), 4.15 (m, 2H), 3.40 (m, 4H), 3.21 (m, 4H), 2.78 (m, 1H), 2.68 (m, 3H), 2.53 (s, 3H), 2.50–2.24 (m, 13H), 2.23 (s, 3H), 2.15–2.05 (m, 5H), 2.05–1.20 (m, 55H), 1.15–0.98 (m, 10H), 0.95 (s, 6H), 0.77 (s, 6H). ^13^C NMR (125 MHz, CD_3_OD) δ: 175.88, 175.58, 174.13, 173.89, 173.67, 172.15, 150.00, 138.20, 136.35, 133.43, 132.09, 123.14, 81.23, 79.82, 76.84, 52.21, 48.67, 48.50, 45.88, 42.29, 41.86, 39.61, 39.17, 38.45, 36.70, 35.74, 35.45, 35.03, 34.28, 34.15, 32.78, 32.44, 31.86, 31.28, 31.07, 30.36, 29.32, 28.57, 27.61, 27.44, 27.15, 26.90, 26.32, 25.94, 25.02, 24.60, 24.50, 24.16, 22.48, 21.73, 21.41, 18.92, 16.92, 16.87, 11.50.

*Molecular umbrella–cispentacin conjugate* **24d**, Starting from 1.02 g (1.44 mmol) of carboxylic acid **19c**, 824 mg (0.39 mmol, 27%) of conjugate **24d** was obtained as a white solid with R_f_ 0.25 (CHCl_3_/MeOH/H_2_O, 7/4/1, *v*/*v*/*v*). ^1^H NMR (500 MHz, CD_3_OD) δ: 6.86 (s, 1H), 6.60 (s, 1H), 4.68 (s, 2H), 4.47 (s, 2H), 4.28 (m, 1H), 4.15 (m, 2H), 3.40 (m, 4H), 3.18 (m, 4H), 2.94–2.61 (m, 4H), 2.55 (s, 3H), 2.49–2.27 (m, 12H), 2.24 (s, 3H), 2.11 (m, 11H), 2.03–1.19 (m, 50H), 1.07 (m, 10H), 0.95 (m, 6H), 0.77 (m, 6H). ^13^C NMR (125 MHz, CD_3_OD) δ: 175.93, 175.65, 174.17, 173.90, 164.92, 164.40, 151.30, 138.08, 135.99, 134.83, 133.28, 123.16, 81.19, 79.94, 76.81, 53.22, 48.54, 46.09, 45.80, 44.05, 42.15, 39.54, 39.33, 38.42, 36.58, 35.57, 35.46, 35.01, 34.32, 34.06, 32.61, 32.44, 31.86, 31.28, 31.10, 30.34, 30.04, 28.55, 27.41, 27.26, 26.28, 25.92, 25.14, 24.49, 24.13, 23.51, 23.71, 22.99, 21.93, 21.92, 21.46, 18.97, 16.51, 11.50.

*Molecular umbrella–Lys(Mca) conjugate* **26a**, Starting from 292 mg (0.45 mmol) of carboxylic acid **19a**, 210 mg (0.13 mmol, 29%) of conjugate **26a** was obtained. The product was purified by liquid column chromatography using a mixtures of solvents CHCl_3_/MeOH/H_2_O, 65/10/1, *v*/*v*/*v* and CHCl_3_/MeOH/H_2_O, 7/4/1, *v*/*v*/*v* as mobile phases. HRMS-ESI: *m*/*z* calcd. for C_90_H_135_N_5_O_18_ 1573.9802; found 1574.9853 [M + 1]^+^. ^1^H NMR (500 MHz, CD_3_OD) δ: 7.68 (d, *J* = 8.9 Hz, 1H), 6.96 (m, 1H), 6.91 (s, 1H), 6.81 (s, 1H), 6.63 (s, 1H), 6.26 (s, 1H), 4.17 (m, 1H), 3.94 (s, 2H), 3.88 (s, 3H), 3.79 (s, 2H), 3.74 (s, 2H), 3.36 (m, 6H), 3.24–3.08 (m, 6H), 2.91 (m, 2H), 2.81 (m, 3H), 2.55 (s, 4H), 2.27 (m, 6H), 2.18–1.19 (m, 57H), 1.18–0.85 (m, 18H), 0.86 (s, 6H). ^13^C NMR (125 MHz, CD_3_OD) δ: 175.37, 172.99, 171.95, 169.04, 163.16, 161.67, 155.34, 151.13, 150.04, 138.09, 136.01, 133.65, 131.91, 126.05, 122.81, 112.68, 100.47, 78.09, 72.51, 71.42, 67.55, 55.18, 54.73, 46.57, 46.02, 41.87, 41.46, 39.57, 39.16, 39.03, 38.83, 36.41, 35.56, 35.09, 34.51, 32.77, 31.92, 31.18, 30.00, 29.75, 29.31, 28.45, 28.21, 27.57, 27.39, 27.19, 26.47, 26.31, 22.57, 24.52, 22.83, 22.45, 21.85, 19.01, 16.37, 11.66.

*Molecular umbrella–Lys(Mca) conjugate* **26b**, Starting from 221 mg (0.34 mmol) of carboxylic acid **19a**, 219 mg (0.10 mmol, 29%) of conjugate **26b** was obtained. The product was purified by liquid column chromatography using a mixture of solvents CHCl_3_/MeOH/H_2_O, 7/4/1, *v*/*v*/*v* as a mobile phase. ^1^H NMR (500 MHz, CD_3_OD) δ: 7.72 (d, *J* = 8.9 Hz, 1H), 6.98 (m, 1H), 6.91 (s, 1H), 6.79 (s, 1H), 6.65 (s, 1H), 6.29 (s, 1H), 4.66 (s, 2H), 4.44 (s, 2H), 4.13 (m, 3H), 3.90 (s, 3H), 3.77 (s, 2H), 3.40 (m, 4H), 3.19 (m, 6H), 2.90 (m, 2H), 2.82 (m, 3H), 2.56–1.18 (m, 67H), 1.18–0.87 (m, 18H), 0.74 (m, 6H). ^13^C NMR (125 MHz, CD_3_OD) δ: 175.48, 173.21, 172.06, 169.00, 163.50, 163.19, 161.74, 155.17, 151.35, 149.91, 137.98, 136.08, 133.56, 131.87, 126.15, 122.85, 112.63, 112.39, 112.24, 100. 56, 81.01, 79.86, 76.43, 55.10, 53.75, 48.39, 46.03, 45.16, 43.45, 42.37, 42.30, 41.94, 39.51, 39.23, 38.73, 38.36, 36.44, 34.57, 34.93, 34.08, 32.78, 31.64, 31.16, 30.33, 30.22, 30.09, 28.96, 28.53, 28.19, 27.61, 27.46, 27.31, 27.18, 26.27, 25.66, 24.64, 24.47, 22.47, 21.77, 18.95, 16.85, 11.53.

*Molecular umbrella–Nap–NH*_2_*conjugate* **27**, Starting from 580 mg (0.82 mmol) of carboxylic acid **19c**, 900 mg (0.41 mmol, 50%) of fluorescent conjugate **27** was obtained as an orange solid with R_f_ 0.29 (CHCl_3_/MeOH/H_2_O, 7/4/1, *v*/*v*/*v*). ^1^H NMR (500 MHz, CD_3_OD) δ: 8.52 (d, *J* = 7.5 Hz, 1H), 8.39 (dd, *J* = 8.2 Hz, 3.1 Hz, 1H), 8.35 (d, *J* = 8.7 Hz, 1H), 8.07 (t, *J* = 5.5 Hz, 1H), 7.98 (m, 1H), 7.76 (m, 1H), 7.64 (t, *J* = 7.9 Hz, 1H), 6.71 (d, *J* = 8.9 Hz, 1H), 6.61 (m, 1H), 6.38 (s, 1H), 4.66 (m, 2H), 4.62 (brs, 1H), 4.44 (m, 2H), 4.13 (m, 4H), 3.53 (m, 2H), 3.19 (m, 4H), 2.72 (m, 2H), 2.53 (m, 2H), 2.44 (m, 4H), 2.35 (m, 9H), 2.22 (m, 2H), 2.12 (m, 6H), 1.98 (m, 7H), 1.92–1.30 (m, 47H), 1.23 (m, 7H), 1.14–0.96 (m, 13H), 0.92 (m, 6H), 0.73 (m, 6H). ^13^C NMR (125 MHz, CD_3_OD) δ: 175.93, 175.65, 174.17, 173.90, 164.92, 164.40, 151.30, 149.75, 138.08, 135.99, 134.83, 133.28, 131.93, 131.07, 129.63, 128.56, 124.44, 122.89, 121.68,120.27, 107.92, 103.66, 81.19, 79.94, 76.81, 60.97, 48.54, 46.09, 45.80, 44.05, 42.15, 41.74, 39.54, 39.33, 39.15, 38.42, 37.75, 36.58, 35.57, 35.46, 35.01, 34.32, 34.06, 32.61, 31.77, 31.10, 30.34, 30.04, 28.55, 27.41, 27.26, 26.28, 25.92, 25.14, 24.49, 24.13, 22.51, 21.71, 19.99, 18.93, 16.92, 15.46, 12.97, 11.51.

*{N*,*N′-bis[4-(tert-butoxycarbonyl)amino]butyl-N*,*N′-bis[3-(tert-butylcarbonyl)amino]propyl}-3,3′-**dithiodipropanoic amide* **29**, To a suspension of 1.76 g (4.34 mmol) of diester **28** in 50 mL of THF, 2 mL (8.68 mmol) of DIPEA and 3 g (8.68 mmol) of protected spermidine **16** was added, respectively. The mixture was stirred at room temperature for 24 h, and then, the solvents were evaporated under reduced pressure. The residue was dissolved in 75 mL of CHCl_3_ and washed with a saturated solution of NaHCO_3(aq)_ (2 × 50 mL), brine (2 × 50 mL), 5% solution of NaHSO_4(aq)_ (2 × 50 mL), and brine (2 × 50 mL), respectively. The organic layer was dried over anhydrous MgSO_4_, the desiccant was filtered off, and the filtrate was concentrated under reduced pressure. The residue was purified by liquid column chromatography using a mixture of solvents CHCl_3_/MeOH/H_2_O, 65/10/11, *v*/*v*/*v* as a mobile phase, obtaining 2.89 g (3.34 mmol, 77%) of product **29** as a colorless oil with R_f_ 0.70 (CHCl_3_/MeOH/H_2_O, 65/10/11, *v*/*v*/*v*). HRMS-ESI: *m*/*z* calcd. for C_40_H_76_N_6_O_10_S_2_ 864.5064; found 865.5112 [M + 1]^+^. ^1^H NMR (500 MHz, CDCl_3_) δ: 3.52–3.22 (m, 8H), 3.20–3.01 (m, 8H), 2.96 (m, 4H), 2.74 (m, 4H), 1.83–1.31 (m, 48H). ^13^C NMR (125 MHz, CDCl_3_) δ: 171.16, 170.43, 156.09, 79.21, 78.88, 49.02, 47.48, 45.66, 45.49, 42.73, 40.11, 39.89, 38.01, 37.29, 34.03, 33.88, 32.89, 37.78, 32.63, 32.52, 29.99, 28.47, 28.43, 27.94, 27.67, 27.49, 26.16, 25.66, 24.94.

*N*^1^,*N*^7^*-bis(tert-butoxycarbonyl)-N*^3^*-(3-thiopropanoyl)spermidine* **30**, To a solution of 2.4 g (2.78 mmol) of spermidine derivative **29** in 50 mL of MeOH, 1.20 g (4.18 mmol) of TCEP solution in 10 mL of water, adjusted to pH 7 with NaHCO_3_, was added. The mixture was stirred at room temperature for 1 h, and then, MeOH was evaporated under reduced pressure. The aqueous residue was extracted with CHCl_3_ (3 × 25 mL). The organic layer was dried over anhydrous MgSO_4_, the desiccant was filtered off, and the filtrate was concentrated under reduced pressure. The residue was purified by liquid column chromatography using a mixture of solvents hexanes/AcOEt, 3/7, *v*/*v* as a mobile phase obtaining 1.72 g (4 mmol, 72%) of thiol **30** as a colorless oil, with R_f_ 0.55 (hexanes/AcOEt, 3/7, *v*/*v*). HRMS-ESI: *m*/*z* calcd. for C_20_H_39_N_3_O_5_S 433.2610; found 434.2623 [M + 1]^+^. ^1^H NMR (500 MHz, CDCl_3_) δ: 6.95–6.63 (m, 2H), 3.21 (m, 4H), 2.91 (m, 4H), 2.63 (m, 4H), 2.29 (m, 1H), 1.77–1.22 (m, 24 H). ^13^C NMR (125 MHz, CDCl_3_) δ: 170.48, 170.23, 156.10, 79.69, 77.89, 60.28, 47.25, 45.20, 45.00, 43.27, 38.13, 36.78, 29.28, 28.73, 28.26, 27.58, 27.23, 26.30, 25.20, 20.39.

*N*^1^,*N*^7^*-bis(tert-butoxycarbonyl)-N*^3^*-[3-(o-hydroxymethylphenyl)dithio]propanoyl]spermidine* **32**, First, 1.56 g (3.60 mmol) of thiol **30** was dissolved in 20 mL of dry MeOH, and then, 1 mL (7.42 mmol) of TEA and 995 mg (4.32 mmol) of activated disulfide **31** were added, respectively. The mixture was stirred at room temperature for 2 h, and then, the solvents were evaporated under reduced pressure. The residue was dissolved in 50 mL of CHCl_3_ and washed with a saturated solution of NaHCO_3(aq)_ (2 × 50 mL), water (2 × 50 mL), 5% solution of NaHSO_4(aq)_ (2 × 50 mL), and water (2 × 50 mL), respectively. The organic layer was dried over anhydrous MgSO_4_, the desiccant was filtered off, and the filtrate was concentrated under reduced pressure. The residue was purified by liquid column chromatography, using a mixture of solvents hexanes/AcOEt, 3/7, *v*/*v* as a mobile phase, obtaining 610 mg (1.07 mmol, 30%) of disulfide **32** as a colorless oil, with R_f_ 0.50 (hexanes/AcOEt, 3/7, *v*/*v*). HRMS-ESI: *m*/*z* calcd. for C_27_H_45_N_3_O_6_S_2_ 571.2750; found 572.2698 [M + 1]^+^. ^1^H NMR (500 MHz, DMSO-*d*_6_) δ: 7.69 (m, 1H), 7.46 (d, *J* = 7.2 Hz, 1H), 7.30 (m, 2H), 6.93–6.66 (m, 2H), 5.31 (t, *J* = 5.4 Hz, 1H), 4.59 (d, *J* = 5.4 Hz, 2H), 3.24–2.99 (m, 6H), 2.92–2.83 (m, 4H), 2.65 (m, 2H), 1.59–1.42 (m, 3H), 1.33–1.20 (m, 3H), 1.37 (s, 18H). ^13^C NMR (125 MHz, DMSO-*d*_6_) δ: 170.96, 170.31, 156.08, 140.30, 135.26, 128.34, 127.53, 78.96, 70.75, 62.74, 60.52, 47.36, 45.42, 42.78, 39.98, 33.91, 32.20, 28.47, 27.55, 26.46, 26.12, 24.97, 21.11, 14.16.

*Molecular umbrella–o-dithobenzoylcarbamoyl linker–cargo conjugates* (**35** and **37**)—*General Procedure*, To a solution of 1.05 mmol of disulfide **32** in 10 mL of MeCN, 1.05 mmol of TEA and 1.05 mmol of DSC were added, respectively. The mixture was stirred at room temperature for 2 h, and then, solvents were evaporated under reduced pressure. The residue was dissolved in 50 mL of AcOEt and washed with water (2 × 20 mL). The organic layer was dried over anhydrous MgSO_4_, the desiccant was filtered off, and the filtrate was concentrated under reduced pressure. The residue was dissolved in 10 mL of MeCN and then transferred to the mixture of 1.58 mmol of a cargo molecule and 2.10 mmol of TEA dissolved in 10 mL of water. The mixture was stirred at room temperature for 2 h, and then, MeCN was evaporated under reduced pressure. The remaining aqueous residue was acidified with 1M HCl_(aq)_ to pH 2, and the resulting solution was extracted with AcOEt (4 × 20 mL). The organic layer was dried over anhydrous MgSO_4_, the desiccant was filtered off, and the filtrate was concentrated under reduced pressure. The residue was roughly purified by liquid column chromatography, using a mixture of solvents CHCl_3_/MeOH/H_2_O, 65/10/1, *v*/*v*/*v* as a mobile phase, obtaining 0.62 mmol of a crude product as an oil. The resulting oil was dissolved in 8 mL of DCM/TFA, 3/1, *v*/*v* mixture and was allowed to stir at room temperature for 1 h. Subsequently, the solvents were evaporated, and the residue was dissolved in 20 mL of dry DMF. Then, 220 μ of DIPEA and 1.24 mmol of **22b** were added, respectively. The mixture was stirred at room temperature for 5 h, and then, most of the solvents were evaporated under reduced pressure, and MeCN was added to the residue. The precipitate was collected and purified by liquid column chromatography using a mixture of solvents CHCl_3_/MeOH/H_2_O, 5/4/1, *v*/*v*/*v* as a mobile phase.

*Molecular umbrella–o-dithobenzoylcarbamoyl linker–cispentacin conjugate* **35**, Starting from 600 mg (1.05 mmol) of disulfide **32**, 230 mg (0.12 mmol, 11%) of conjugate **35** was obtained as a beige solid, with R_f_ 0.30 (CHCl_3_/MeOH/H_2_O, 5/4/1, *v*/*v*/*v*). ^1^H NMR (500 MHz, CD_3_OD) δ: 8.00 (m, 2H), 7.80 (m, 1H), 7.52–7.37 (m, 2H), 7.32 (m, *J* = 6.1 Hz, 1H), 5.41–5.17 (m, 2H), 4.66 (s, 2H), 4.46 (s, 2H), 4.16 (m, 3H), 3.35 (m, 2H), 3.18 (m, 6H), 3.01 (m, 2H), 2.89 (m, 1H), 2.75 (t, 2H), 2.49–2.22 (m, 10H), 2.10 (m, 6H), 2.04–1.20 (m, 42H), 1.17–0.98 (m, 10H), 0.95 (s, 6H), 0.76 (s, 6H). ^13^C NMR (125 MHz, CD_3_OD) δ: 175.74, 164.79, 129.80, 128.70, 127.30, 125.61, 109.81, 81.06, 79.81, 76.60, 63.79, 54.28, 48.47, 46.06, 45.96, 45.38, 42.22, 42.04, 39.31, 35.68, 35.47, 35.11, 34.43, 34.11, 32.71, 31.99, 31.80, 31.62, 30.30, 29.40, 28.94, 28.29, 27.61, 27.43, 27.35, 27.15, 26.37, 25.80, 24.60, 22.52, 21.79, 16.89, 11.56.

*Molecular umbrella–o-dithobenzoylcarbamoyl linker–Lys(Mca) conjugate* **37**, Starting from 630 mg (1.10 mmol) of disulfide **32**, 230 mg (0.11 mmol, 10%) of conjugate **37** was obtained as a light-yellow solid, with R_f_ 0.50 (CHCl_3_/MeOH/H_2_O, 5/4/1, *v*/*v*/*v*). ^1^H NMR (500 MHz, CD_3_OD) δ: 8.82 (s, 1H), 7.78 (m, 2H), 7.42 (d, *J* = 7.5 Hz, 1H), 7.38 (t, *J* = 7.3 Hz, 1H), 7.30 (t, *J* = 7.6 Hz, 1H), 7.04 (m, 1H), 6.98 (m, 1H), 5.24 (m, 2H), 4.65 (m, 2H), 4.46 (m, 2H), 4.16 (m, 2H), 3.96 (m, 3H), 3.51 (m, 2H), 3.37 (m, 2H), 3.30–3.12 (m, 8H), 3.01 (m, 2H), 2.78 (m, 2H), 2.47–2.20 (m, 10H), 2.20–1.18 (m, 42H), 1.14–0.84 (m, 16H), 0.74 (m, 6H).

### 4.2. Microbial Strains and Culture Conditions

The reference strains used in this study were *Candida albicans* ATCC 10231, *Candida glabrata* DSM 11226, and *Candida krusei* DSM 6128. *C. albicans* B3, B4, Gu4, and Gu5 clinical isolates were kindly provided by Joachim Morschhäuser, Würzburg, Germany. Gu4 and B3 are fluconazole-sensitive isolates obtained from early infection episodes, while Gu5 and B4 are the corresponding fluconazole-resistant isolates obtained from later episodes in the same patients treated with fluconazole [24]. Strains were grown at 30 °C in YPD medium (2% glucose, 1% yeast extract, and 1% Bacto-Peptone) and stored on YPD agar plates containing 2% agar.

### 4.3. Antifungal In Vitro Activity Determination

Susceptibility testing was performed in two growth media: (a) RPMI-1640 *w*/*o* sodium bicarbonate, with L-glutamine + 2% glucose + 3.45% MOPS, pH adjusted to 7.0; (b) YNB-AS—Yeast Nitrogen Base *w*/*o* amino acids (contains ammonium sulfate 5 g L^−1^) + 2% glucose. The in vitro growth inhibitory activity of antifungals was quantified by determination of MIC_50_ and MIC_90_ values by the serial two-fold dilution method, using the 96-well microtiter plates. Serial dilutions of cispentacin and its conjugates were prepared in the 4–0.015 mM range.

Conditions of the RPMI-1640-based assay were the same as outlined in the CLSI recommendations [25] except for the end-point readout that was done by spectrophotometric determination of cell density at 660 nm. Turbidity in individual wells was measured with a microplate reader (Victor^3^; Perkin Elmer). The 96-well microtiter plates were also used for the determination of in vitro growth inhibitory activity in YNB-AS medium. Individual wells were inoculated with 5 × 10^3^ *Candida* cells mL^−1^ from the overnight culture in YPD medium. The plates were incubated at 37 °C for 24 h, and then, turbidity was measured with a microplate reader at 660 nm, as described above for the RPMI-1640-based assay.

Values of MIC_50_ and MIC_90_ were read from the graphs of A_660_ vs. drug concentration and corresponded to the interpolated values of drug concentration at which A_660_ was 50% or 10% of that measured for the drug-free control.

### 4.4. Hemolytic Activity Determination

Red blood cell concentrates were kindly provided by the Regional Center for Blood Donation and Blood Treatment in Gdańsk. The hemolytic activity determination was carried out by the serial dilution method, according to the procedure described earlier [26]. Briefly, human erythrocytes were suspended in saline to give a suspension of 2 × 10^7^ cells mL^−1^ (hemocytometer count). The stock 1 mg mL^−1^ solutions of conjugates were prepared in DMSO, and 50 μL aliquots of serial two-fold dilutions were placed in Eppendorf tubes. Tubes containing 50 μL of DMSO and 50 μL of 2% aqueous Triton X-100 solution were included as a negative and positive control, respectively. To each tube, 950 μL of the erythrocyte suspension was added and mixed by inversion to give the final concentrations of compounds tested in the 200–0.4 μg mL^−1^ range. The samples were incubated at 37 °C for 30 min; then, they were mixed by inversion and centrifuged (1700× *g*, 5 min, 4 °C). The concentration of hemoglobin in supernatants obtained after the centrifugation of erythrocytes suspension was determined by measuring the absorbance at wavelength λ = 540 nm (A_540_^sample^). Absorbance of the negative (A_540_^DMSO^) and the positive (A_540_^0.1%Triton X−100^) controls was also measured. The percent of hemolysis at a given compound concentration/EH (%)/was calculated as follows:EH (%) = ((A_540_^sample^ − A_540_^DMSO^)/(A_540_^0.1%Triton X−100^ − A_540_^DMSO^)) × 100.

The EH_50_ values for each compound were calculated with the GraphPad Prism software as an interpolated concentration of compound, for which the A_540_ value is exactly 50% of the A_540_ value measured for the positive control sample.

### 4.5. Preparation of C. albicans Cell-Free Extracts

*C. albicans* cells were harvested from the culture in YPD medium in the logarithmic phase of growth and washed with PBS, pH 7.4. Then, cells were suspended in a minimal amount of PBS to an optical density OD_660_ ≈ 1.5. A sample of 1.5 mL of the cell suspension was put into the Lysing Matrix D, 2 mL tubes containing 1.4 mm ceramic spheres (MP Biomedicals). The tubes were vortexed 5 times for 2 min intervals, with 2 min breaks for cooling in the ice/water bath. Finally, the mixtures were centrifuged (10,000× *g*, 4 °C, 20 min), and the supernatant was pushed through the syringe filter, 0.22 μm pore size, to remove unbroken cells. The thus-obtained cell-free extract was kept at 4 °C and used no later than 5 h after preparation.

### 4.6. HPLC-MS Analysis of Conjugate Cleavage

Analysis of cleavage of the conjugates was performed in two experimental systems: (A) model system, in the presence of pig liver esterase or glutathione, reduced; (B) in *C. albicans* cell-free extract.

System A

Mixtures containing a given conjugate at the initial concentration of 150 µM and pig liver esterase, 5 mg mL^−1^ or 10 mM glutathione, reduced, in a total volume of 200 μL were incubated at 30 °C. Samples of 50 μL were collected at zero time, after 30 min, and after 120 min and subjected to HPLC-MS analysis.

System B

A given conjugate was dissolved in 4 mL of *C. albicans* cell-free extract to the initial concentration of 150 µM. Mixtures were incubated at 30 °C. Samples of 1 mL were collected at zero time, after 30 min, and after 120 min. The collected samples were immediately de-proteinized by centrifugation (4 °C, 5000× *g*, 10 min) in Amicon^®^ Ultra-4 Centrifugal Filter Units, 3 kDa cutoff limit. Samples of 50 μL were collected from the filtrates and subjected to HPLC-MS analysis.

The HPLC-MS system consisted of a liquid chromatograph, a degasser, a binary pomp, an auto-sampler, and a column oven, which was combined with an MS detector with an electrospray source (AJS ESI) and quadrupole analyzer (1260 Infinity II and 6470 Triple Quad LC/MS, Agilent Technologies, Waldbronn, Germany). The ChemStation software was used to control the LC-MS system and for data processing.

Chromatographic separations were performed on an Eclipse XDB-C18 column (150 mm × 4.6 mm, 5 μm). For the separation, a gradient of mobile phase A (H_2_O + 0.01% HCOOH) and mobile phase B (100% methanol) was used. The gradient profile was set as follows: 0 min—95% effluent A, 8 min—70% effluent A, 11 min—50% effluent A, 12 min—95% effluent A, 17 min—95% effluent A. The flow rate was 1 mL min^−1^, the column temperature was 25 °C, and the injection volume was 20 μL.

The electrospray source operated in a negative mode. The data were collected in a MS scanning mode (MS2 SCAN) with the range 150–700 (*m*/*z*).

### 4.7. Microscopic Examination of Conjugate Uptake

*Candida* cells from the overnight culture in YPD medium were harvested, washed with distilled water, and used for inoculation of RPMI-1640 medium. The culture was grown at 30 °C with shaking from OD_660_ ≈ 0.1 to OD_660_ ≈ 0.3. The cells were harvested by centrifugation and immediately suspended in PBS to OD_660_ ≈ 0.2. A conjugate containing fluorescent probe was added to the final concentration of 50 µg mL^−1^, and the cell suspension was incubated (30 °C, 150 rpm). Samples of 2 mL were collected at zero time and at time intervals, centrifuged, and washed 4 times with PBS. After the final wash, the cells were suspended in a small volume of PBS and immediately observed using the confocal microscopy (63× magnification; ZEISS LSM T-PMT, Magdeburg, Germany). The imaging conditions were as follows: conjugates containing Lys(Mca)—excitation 350 nm, emission 385 nm, and conjugates containing Nap-NH_2_—excitation 438 nm, emission 527 nm.

## Data Availability

For the purpose of Open Access, the author has applied a CC-BY public copyright license to any AAM version arising from this submission. Publicly available datasets were analyzed in this study. The data presented in this study are openly available from 4 August 2021 in [https://mostwiedzy.pl/pl/open-research-data] at [doi:10.34808/m04n-wx77].

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
