# Peer review of "Molecular Umbrella as a Nanocarrier for Antifungals"

_molecules, 2021, doi:10.3390/molecules26185475_

Round 1

Reviewer 1 Report

The paper reported some molecular umbrella as antifungals, it is interesting and can be accepted after minor revision. The introduction should be added more content. In this paper, only 7 references had been cited in the introduction. It is not enough.

Author Response

The paper reported some molecular umbrella as antifungals, it is interesting and can be accepted after minor revision. The introduction should be added more content. In this paper, only 7 references had been cited in the introduction. It is not enough.

Authors' response

Four further references have been added to Introduction. Now, in Introduction, there are 15 references cites (please note that at some reference numbers there are more than one reference.

Reviewer 2 Report

Review Report for The Authors

Manuscript Number: Molecules 2021, 26, x FOR PEER REVIEW

This manuscript describes the “Molecular umbrella as a nanocarrier for antifungals”. The design and execution of biological assays seem significantly considerable”. The manuscript highlights the Molecular umbrella composed of two O-sulfated cholic acid residues was applied for the construction of conjugates with cispentacin, containing “trimethyl lock” (TML) or o-dithiobenzyl-carbamoyl moiety as a cleavable linker. Three out of five conjugates demonstrated antifungal in vitro activity against C. albicans and C. glabrata but not against C. krusei, with MIC90 values in the 0.22 –0.99 mM range and were not hemolytic. Also stated that the Antifungal activity of the most active conjugate 24c, containing the TML-pimelate linker, was comparable to that of intact cispentacin. Structural analogue of 24c, containing the Nap-NH2 fluorescent probe, was accumulated in Candida cells and TML containing conjugates were cleaved in cell-free extract of C. albicans cells.

“Although, results suggest that molecular umbrella can be successfully applied as a nanocarrier for the construction of cleavable antifungal conjugates” The Manuscript and the studies are impressive and attractive for the readers in the field of antimicrobial drug discovery. It is indeed an important goal to discover highly effective antifungal agents. Nevertheless, there are several minor issues with this paper that preclude its publication at this time. details that need to be corrected are listed below.

Minor concerns:

  • Initially, the authors need to incorporate the chemical structures of 1) Cispentacin, a new antifungal antibiotic 2) Lys(Mca) and 3) Nap-NH2 as a figure 1A, under the section called Rationale for design of conjugates.
  • All the synthetic scheme legends essentially need to be changed in simple and straight forward fashion. Ex: DMAP, DCM, 0°C --> rt;? Please find the reference given below and suggested to cite them.
  • Throughout the MS the chemistry experimental data has some errors in the representation. Like, there are no coupling constants (J) reported for any compounds in the NMR data. Which is not possible for any compound to be published without J values at d or t? The authors must require to be added the J

J-couplings (also called spin-spin coupling or indirect dipole–dipole coupling)

  • Suggesting to refer previous published journal articles (given below) for writing NMR data and correct it. When 1HNMR taken at 500 MHz, the 13CNMR should be recorded technically at 125 MHz not at same 500 MHz?
  • Suggested to remove the word analog, because they are all conjugates.
  • The compound synthesis description missed the references? Authors need to add related references in the text.The authors need to add update all the references in order to meet recent literature in the specific field of studies.
  • In the text throughout the manuscript spaces, time as (h), misspelled (eg, NMR data representation, qt not qi). ??

The references to be reviewed for writing the NMR data.

  • Sharma KK, et al., Discovery of a Membrane-Active, Ring-Modified Histidine Containing Ultrashort Amphiphilic Peptide That Exhibits Potent Inhibition of Cryptococcus neoformans. J Med Chem. 2017 Aug 10;60(15):6607-6621. PMID: 28697301.
  • Kamal A, et al., Synthesis, biological evaluation of new oxazolidino-sulfonamides as potential antimicrobial agents. Eur J Med Chem. 2013 Apr;62:661-9. PMID: 23434639.
  • Sharma KK, et al., Modified histidine containing amphipathic ultrashort antifungal peptide, His[2-p-(n-butyl)phenyl]-Trp-Arg-OMe exhibits potent anticryptococcal activity. Eur J Med Chem. 2021, Jun 12; 223:113635. PMID: 34147743.
  • Al-Hiari, et al., Synthesis and Antibacterial Properties of New 8-Nitrofluoroquinolone Derivatives. Molecules 2007, 12, 1240-1258

I would recommend the authors address all the issues mentioned, with that the manuscript quality considerably enhanced.

Overall, the studies seem excellent to be considerable and appreciated their efforts to conduct such experiments. Still, there are many shortcomings that will preclude its publication at this time.

Author Response

Initially, the authors need to incorporate the chemical structures of 1) Cispentacin, a new antifungal antibiotic 2) Lys(Mca) and 3) Nap-NH2 as a figure 1A, under the section called Rationale for design of conjugates

Authors' response

In fact, cispentacin is not a new antibiotic. It was found in the nineties of the previous century. The structure of this agent, as well as these of  are clearly shown in many Schemes and Figures, for the first time in Figure 1 as "cargoes". We have added an explanation in the legend.

-------------------

All the synthetic scheme legends essentially need to be changed in simple and straight forward fashion. Ex: DMAP, DCM, 0°C --> rt;? Please find the reference given below and suggested to cite them.

Authors' response

All the scheme legends have been modified, following the reviewer's suggestion

-----------------------------------

  • Throughout the MS the chemistry experimental data has some errors in the representation. Like, there are no coupling constants (J) reported for any compounds in the NMR data. Which is not possible for any compound to be published without J values at d or t? The authors must require to be added the J

J-couplings (also called spin-spin coupling or indirect dipole–dipole coupling)

  • Suggesting to refer previous published journal articles (given below) for writing NMR data and correct it. When 1HNMR taken at 500 MHz, the 13CNMR should be recorded technically at 125 MHz not at same 500 MHz?

Authors' response

All the NMR data have been corrected by adding the coupling constants (J) values and correcting the MHz value for 13C NMR experiments.

-------------------------

Suggested to remove the word analog, because they are all conjugates.

The word "analog" has been replaced in two cases, where it was justified and left unchanged, where appropriate.

----------------------------------------

The compound synthesis description missed the references? Authors need to add related references in the text.The authors need to add update all the references in order to meet recent literature in the specific field of studies.

Authors' response

References are given whereever necessary. In most cases, our synthetic procedures have been significantly modified, so that the whole descriptions are provided instead of giving a reference.

------------------------------------

In the text throughout the manuscript spaces, time as (h), misspelled (eg, NMR data representation, qt not qi).

Authors' response

Corrected appropriatelly.

Reviewer 3 Report

The authors in the present manuscript a series of Molecular umbrella derivatives conjugated with the active anti fugal drug in attempt to form nano carrier to well deliver the drug to the target (host organ or pathogen). The synthesis of the derivatives seems logic and the obtained derivatives were also been determined based on NMR and MS analysis. The in vitro activity and the intracellular distribution were also studied. Several comments here are put forwarded in order to improve in the revised version.

  1. What is meaning for the NMR signal of protons which labelled as qi. There many places in the context.
  2. If the H-1 was measured at 500MHz, the C-13 can only be measured at 125 MHz, please correct all mistakes in the C-13 NMR data. 
  3. The HPLC-MS figs were not clear, please change them. 
  4. Also, although the author mentioned, it is still necessary to determine or discuss the unexpected peak at 145(m/z). It is strange to compare with authentic sample because their same retention time(?).
  5. There are several editing errors, please recheck it.

Author Response

  • What is meaning for the NMR signal of protons which labelled as qi. There many places in the context.
  • If the H-1 was measured at 500MHz, the C-13 can only be measured at 125 MHz, please correct all mistakes in the C-13 NMR data. 

Authors' response

All NMR data have been corrected appropriatelly

---------------------------

The HPLC-MS figs were not clear, please change them. 

Authors' response

The figures have been replaced

---------------------------------------------------

Also, although the author mentioned, it is still necessary to determine or discuss the unexpected peak at 145(m/z). It is strange to compare with authentic sample because their same retention time(?).

Authors' response

This is indeed surprising and so far we have no explanation for this.  We would not like to speculate. So that we have only indicated that the retention time of the m/z 145 peak was similar to that of the m/z 128 peak (not necessarilly identical), although a possible difference is small.

----------------------------------------

There are several editing errors, please recheck it.

Authors' response

A carefull correction has been made.